# The phenuivirus Toscana virus makes an atypical use of vacuolar acidity to enter host cells

Jana Koch[1,2,3], Qilin Xin[3], Martin Obr[4], Alicia Schäfer[1,2], Nina Rolfs[1,2], Holda A. Anagho[1,2], Aiste Kudulyte[1,2], Lea Woltereck[1,2], Susann Kummer[1¤], Joaquin Campos[5], Zina M. Uckeley[1,2], Lesley Bell-Sakyi[6], Hans-Georg Kräusslich[1], Florian KM. Schur[4], Claudio Acuna[5], Pierre-Yves Lozach[1,2,3]*

1 Center for Integrative Infectious Diseases Research (CIID), University Hospital Heidelberg, Heidelberg, Germany, 2 CellNetworks–Cluster of Excellence, Heidelberg, Germany, 3 Univ. Lyon, INRAE, EPHE, IVPC, Lyon, France, 4 Institute of Science and Technology Austria (ISTA), Klosterneuburg, Austria, 5 Chica and Heinz Schaller Research Group, Institute of Anatomy and Cell Biology, Heidelberg University, Heidelberg, Germany, 6 Institute of Infection, Veterinary and Ecological Sciences, University of Liverpool, Liverpool, United-Kingdom

¤ Current Address: Center for Biological Threats and Special Pathogens, Robert Koch Institute, Berlin, Germany
* pierre-yves.lozach@inrae.fr

**Data Availability Statement:** All data are available in the main text or the supplementary materials.

**Funding:** This work was supported by CellNetworks Research Group funds and Deutsche Forschungsgemeinschaft (DFG) funding (LO-2338/

## Abstract

Toscana virus is a major cause of arboviral disease in humans in the Mediterranean basin during summer. However, early virus-host cell interactions and entry mechanisms remain poorly characterized. Investigating iPSC-derived human neurons and cell lines, we found that virus binding to the cell surface was specific, and 50% of bound virions were endocytosed within 10 min. Virions entered Rab5a+ early endosomes and, subsequently, Rab7a+ and LAMP-1+ late endosomal compartments. Penetration required intact late endosomes and occurred within 30 min following internalization. Virus entry relied on vacuolar acidification, with an optimal pH for viral membrane fusion at pH 5.5. The pH threshold increased to 5.8 with longer pre-exposure of virions to the slightly acidic pH in early endosomes. Strikingly, the particles remained infectious after entering late endosomes with a pH below the fusion threshold. Overall, our study establishes Toscana virus as a late-penetrating virus and reveals an atypical use of vacuolar acidity by this virus to enter host cells.

## Author summary

Toscana virus (TOSV) is a reemerging sandfly-borne enveloped virus causing neuro-invasive infections in humans. The virus is endemic in the Mediterranean basin, with a potential risk of introduction in northern Europe and Asia. Despite its significance, diagnostics, therapeutics, and research on TOSV have been neglected. Here, we developed accurate, sensitive methods to examine the early stages of TOSV infection in both fixed and live human neurons and cell lines, covering virus binding to fusion. Our results highlight the crucial role of late endosomal maturation in TOSV entry, shedding light on atypical viral

3-1) and the Agence Nationale de la Recherche (ANR) funding (grant numbers ANR-21-CE11-0012 and ANR-22-CE15-0034), all awarded to P.-Y.L. This work was also supported by the LABEX ECOFECT (ANR-11-LABX-0048) of Université de Lyon (UDL), within the program "Investissements d'Avenir" (ANR-11-IDEX-0007) operated by the ANR and by the RESPOND program of the UDL (awarded to P.-Y.L) . C.A. was supported by the Chica and Heinz Schaller Research Group funds, NARSAD 2019 award, a Fritz Thyssen Research Grant, and the SFB1158-S02 grant. L.B-S. is supported by a United Kingdom Biotechnology and Biological Sciences Research Council grant (BB/P024270/1) and a Wellcome Trust grant (223743/Z/21/Z). F.K.M.S acknowledges support from the Austrian Science Fund (FWF, P31445). J.K. received a salary from the DFG (LO-2338/3-1) and then from the ANR (ANR-11-LABX-0048). The salary of Z.M.U. was partially covered by the DFG (LO-2338/3-1). S.K. received a salary from the DFG (SFB1129). We are grateful to the Chinese Scholarship Council (CSC; 201904910701), DAAD/ANID (57451854/62180003), the Rufus A. Kellogg fellowship program (Amherst College, Massachusetts, USA) for awarding fellowships to Q.X., J.C., and H.A.A., respectively. The funders had no role in study design, data collection and analysis, decision to publish, or preparation of the manuscript."

**Competing interests:** The authors have declared that no competing interests exist.

fusion mechanisms. Notably, the fusion process of TOSV appears, at least in part, reversible, and the virus depends on progressive acidification in endosomes for activation and penetration, rather than a specific pH threshold. The information gained here lays the basis for future research into entry inhibitors against not only TOSV, but all viruses using similar penetration strategies. Our study also emphasizes that only a synergistic combination of innovative structure-function and cell biology analyses will provide a better understanding of virus fusion in the cellular context.

## Introduction

Toscana virus (TOSV) is a re-emerging sand fly-borne pathogen of the *Phenuiviridae* family (genus *Phlebovirus*, order *Bunyavirales*) that is responsible for neuro-invasive infections in humans and causes meningitis and meningoencephalitis in most severe cases [1]. The virus was first isolated from the phlebotomine sand flies *Phlebotomus perniciosus* and *Phlebotomus perfiliewi* in Tuscany, central Italy, in 1971 [2]. Nowadays, TOSV is widely spread in North Africa and southern Europe, including Greece, Italy, southern France, and Spain [1, 3, 4]. TOSV is currently a primary cause of arthropod-borne viral disease in humans in Mediterranean countries during summer [4]. However, until now, no vaccines or antiviral treatments are approved for human use.

TOSV has a tri-segmented, single-stranded RNA genome of predominantly negative polarity that replicates exclusively in the cytosol of infected cells [5]. The larger segment (L) codes for the RNA-dependent RNA polymerase that is required to initiate virus replication after the release of the viral genome into the cytosol. The medium segment (M) encodes a polyprotein precursor, the proteolytic cleavage of which results in a non-structural protein, NSm, and two envelope glycoproteins, Gn and Gc. The smallest genomic segment (S) codes for the non-structural protein NSs, and for the nucleoprotein N which associates with the RNA genome and constitutes, together with the viral polymerase, the ribonucleoproteins (RNPs) [6]. Viral particles are believed to assemble and acquire their lipid envelope in the endoplasmic reticulum or Golgi network, from where newly-formed viral particles leave infected cells.

TOSV has so far not been visualized, and hence the morphology, size, and structural organization of virions remain to be determined. Other phenuiviruses are enveloped, roughly spherical, and about 100 nm in diameter [7]. While viral RNPs are inside the viral particles, the two envelope glycoproteins Gn and Gc decorate the outer surface and allow virus binding to host cells and acid-activated penetration into the cytosol. Cryo-electron tomography revealed that most regular phenuiviral particles have protrusions of about ten nanometers forming an icosahedral lattice with an atypical T = 12 triangulation [8, 9]. Structural studies of Rift Valley fever virus (RVFV), Dabie virus (DABV), and Heartland virus (HRTV) revealed that the phenuiviral Gc belongs to the group of class-II membrane fusion proteins [10–12].

The tropism, receptors, cellular factors, and pathways used by TOSV to enter and infect host cells are largely unidentified and poorly characterized. The virus was shown to subvert heparan sulfates and the C-type lectins DC-SIGN and L-SIGN to attach to the cell surface [13–15]. A few phenuiviruses have been shown to depend on endocytic internalization and vacuolar acidification for infectious entry [7]. As with other class-II fusion proteins, acidic pH is thought to trigger multiple conformational changes in phenuiviral glycoproteins that lead to the insertion of the viral fusion unit into endosomal membranes [16]. Foldback of the viral fusion protein follows and then the formation of a fusion pore allows the release of the virus genome into the cytosol.

Here, we analyzed the entry of TOSV into induced pluripotent stem cell (iPSC)-derived human neurons and other tissue culture cells. To this end, we developed sensitive fluorescence-based approaches to examine and quantify TOSV infection, binding, internalization, intracellular trafficking, and membrane fusion. The results showed that TOSV shares with other phenuiviruses a dependence on the degradative branch of the endocytic machinery for penetration of host cells by acid-activated membrane fusion. We found that TOSV is a class-II fusion virus, but it differs greatly from other viruses in this class in that its fusion process is, at least in part, reversible and its activation depends on the progressive acidification within maturating endosomes, rather than a specific pH threshold. In other words, TOSV made atypical use of endosomal acidity to find its way through endosomal vesicles and enter the cytosol.

## Results

### TOSV infects human iPSC-derived neurons

TOSV causes meningoencephalitis in the most severe cases. Therefore, we sought to test the sensitivity of brain cells to TOSV infection. To this end, functional, human glutamatergic neurons were generated from iPSCs through expression of the transcription factor neurogenin-2 (NGN2) [17] and exposed to different multiplicities of infection (MOIs) of TOSV for 48 h. The susceptibility of cells was assessed by flow cytometric analysis after immunofluorescence staining with antibodies (Abs) directed against all TOSV structural proteins, *i.e.*, the nucleoprotein N and the glycoproteins Gn and Gc. Nearly 70% of the neuronal cells were infected at the highest MOI (Fig 1A). The percentage of infected cells increased over time and reached a plateau within 16 h post-infection (hpi) (Fig 1B), indicating that the fluorescence signal detected in this assay corresponded to viral replication and not to the input virions.

Several cell types of various species have been reported to support productive TOSV infection [5, 14], suggesting broad host range potential and wide tissue tropism of the virus. We evaluated 18 further cell lines from different arthropod and vertebrate species and found that 16 were susceptible and permissive to infection and seven of eight tested cell lines produced infectious viral particles, as determined by our flow cytometric and plaque-forming unit (pfu) assays (S1 Table and Fig 1C and 1D). Of note, myeloid or lymphatic lineages were poorly infected, if at all. The three sand fly cell lines, namely LLE/LULS40, LLE/LULS45, and PPL/LULS49, allowed a complete viral cycle, but their sensitivity to virus infection was low compared to that of most of the mammalian cells.

To evaluate the production and release of infectious viral particles, we infected A549 cells at low MOIs and quantified virus infection and production up to 48 hpi (Figs 1E and 1F and S1). Infectious progeny was found to be released from infected cells as early as 9–16 hpi. Collectively, our analysis indicated that TOSV completes one round of infection, from virus binding and penetration to replication and release of infectious progeny within 9 h in A549 cells. Similar results were obtained in other cell lines tested (S1 Table). As we aimed to analyze TOSV entry mechanisms and restrict infection to a single round in the selected cell lines, we limited our assays to 6 hpi in all further experiments. In addition, we used MOIs for each cell line allowing the infection of approximately 20% of cells. This range of infection generally avoids saturation of infection in cells and thus, allows the detection of potential inhibitory or enhancing effects of a perturbant. Together, this series of data indicates that TOSV efficiently infects human neurons and various mammalian cell lines. In most of the following experiments, we opted to use A549 cells as they are easier to handle than the complex iPSC-derived neuron system.

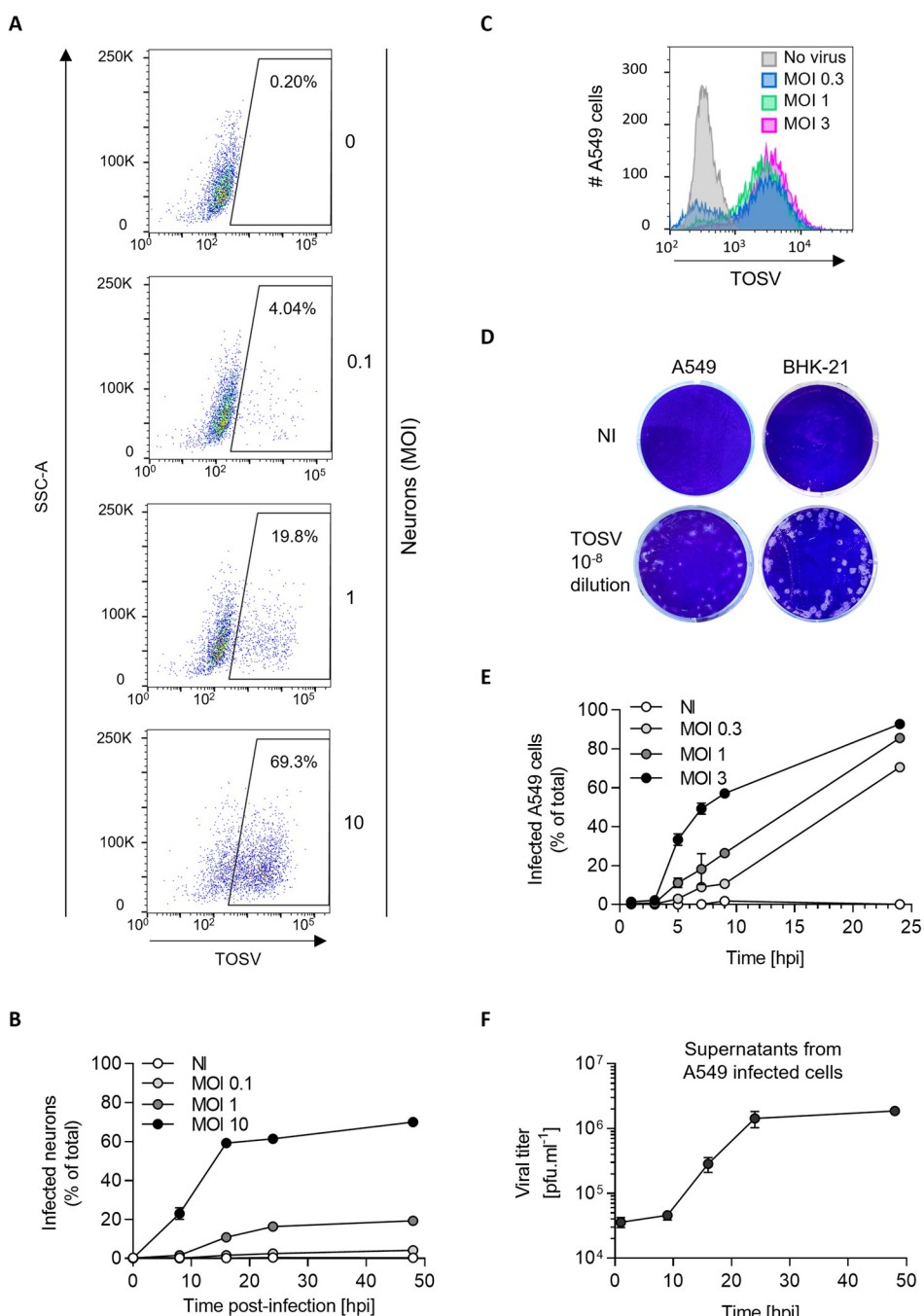

**Fig 1. Toscana virus (TOSV) productively infects iPSC-derived human neurons.** (**A**) Induced human pluripotent stem cell (iPSC)-derived neurons were infected with TOSV at the indicated multiplicities of infection (MOIs). Intracellular expression of viral proteins was detected by immunostaining and flow cytometry analysis 48 h later. (**B**) TOSV infection was monitored in iPSC-derived neurons over 48 h assay as described in A. (**C**) A549 cells were infected with TOSV for 24 h, and the expression of viral proteins was analyzed by immunostaining and flow cytometry. (**D**) The titer of TOSV stocks produced on BHK-21 cells was assessed on A549 cells and BHK-21 cells by a plaque-forming unit (pfu) assay. Examples are shown for $10^{-8}$ dilutions of virus productions. (**E**) TOSV infection was monitored in A549 cells over 24 h and analyzed as described in C and S1 Fig. (**F**) Supernatants from A549 cells challenged with TOSV at an MOI 2 were assessed for the production of infectious viral progeny over 48 h by the pfu assay described in D.

## Labeling of TOSV with fluorescent dyes

To visualize and quantify the early stages of TOSV entry, we purified and labeled the virus chemically with fluorescent probes. Hydroxysuccinimidyl (NHS) ester dyes with an excitation wavelength of 488 nm (Alexa Fluor [AF] 488) or 647 nm (ATTO647N) were coupled to free lysines in the glycoproteins Gn and Gc at a dye-to-glycoprotein ratio between 1:1 and 2:1. At this ratio, we assumed that all virions were labeled. Alternatively, TOSV was labeled with the lipid dye R18 primarily for the analysis of viral fusion. A high concentration of R18 molecules in viral envelopes results in autoquenching of the fluorescence signal [18]. Viral fusion allows the release of R18 into the target cell membranes leading to the dilution of the dye, and, thus, dequenching of the fluorescence.

Characterization of the fluorescent virions is shown in S2 Fig. Briefly, labeled particles were purified through a sucrose gradient so that unbound dyes were removed (S2A Fig). Analysis by SDS-PAGE and Coomassie blue staining showed that the purity of the labeled TOSV preparations was greater than 90% (S2B Fig). The only proteins labeled with NHS ester dyes were Gn and Gc (S2B Fig). The observation that N was not labeled demonstrated that the viral envelope was intact during the labeling procedure. The different labeled particles could be visualized as single spots by confocal microscopy and super-resolution stimulated emission depletion (STED) microscopy (S2C Fig). We did not notice a significant impact of labeling on TOSV infectivity. The titers were similar to those of non-labeled particles (S2D Fig).

TOSV particles were then imaged by cryo-electron microscopy (EM) after fixation with paraformaldehyde and vitrification (Fig 2A). Virions appeared roughly spherical with a diameter of 121 ± 11 nm (n = 96) and protrusions of 9 ± 2 nm (n = 96) (Fig 2B). The measured roundness coefficient of virions was close to 1, *i.e.*, the ratio between their perpendicular width and length was of 0.9 ± 0.1 (n = 96) (Fig 2B). This reflected the nearly spherical shape of the viral particles. Overall, TOSV particles displayed the typical morphology known for other phenuiviruses such as RVFV and Uukuniemi virus (UUKV) [8, 9].

## TOSV binding to cells is specific

To evaluate TOSV binding to cells, R18-labeled virions were first allowed to bind to A549 cells in suspension at 4˚C for 90 min. Incubation on ice allows virions to bind to the cell surface but prevents their internalization into cells. The unbound virions were washed away, and the total fluorescence associated with cell-bound viral particles was determined using a fluorimeter. The results showed that the fraction associated with the cells was about 25% of the virus input (Fig 2C). After binding of ATTO647N-labeled particles to A549 cells at 4˚C, virus particles were imaged by confocal microscopy and could be detected on the cell surface (Fig 2D). That the spots had varied sizes suggested that not only individual virions were attached to cells. The largest clusters were probably formed by 2–3 virions but were only seen as a single spot due to the limitation of confocal resolution. The fact that the number of spots per cell was 10.5 ± 8.5 (n = 151) at MOI only 1, *i.e.*, one infectious virion initially added per cell, indicated that the ratio between infectious and noninfectious bound virions was around 1:10.

Flow cytometry analysis allowed detection and quantification of TOSV-AF488 from MOI 3 and above (Fig 2E). Binding of TOSV-AF488 was observed to be abrogated by pre-binding of non-labeled TOSV in a dose-dependent manner (Fig 2F). A 33-fold higher concentration of non-labeled virions reduced TOSV-AF488 binding by one-fourth. Complete inhibition was not achieved because the necessary concentrations of the non-labeled virus were not reached under our experimental settings. However, the pre-binding of Semliki Forest virus (SFV) did not affect TOSV-AF488 binding under the same conditions. Combined, the data indicated

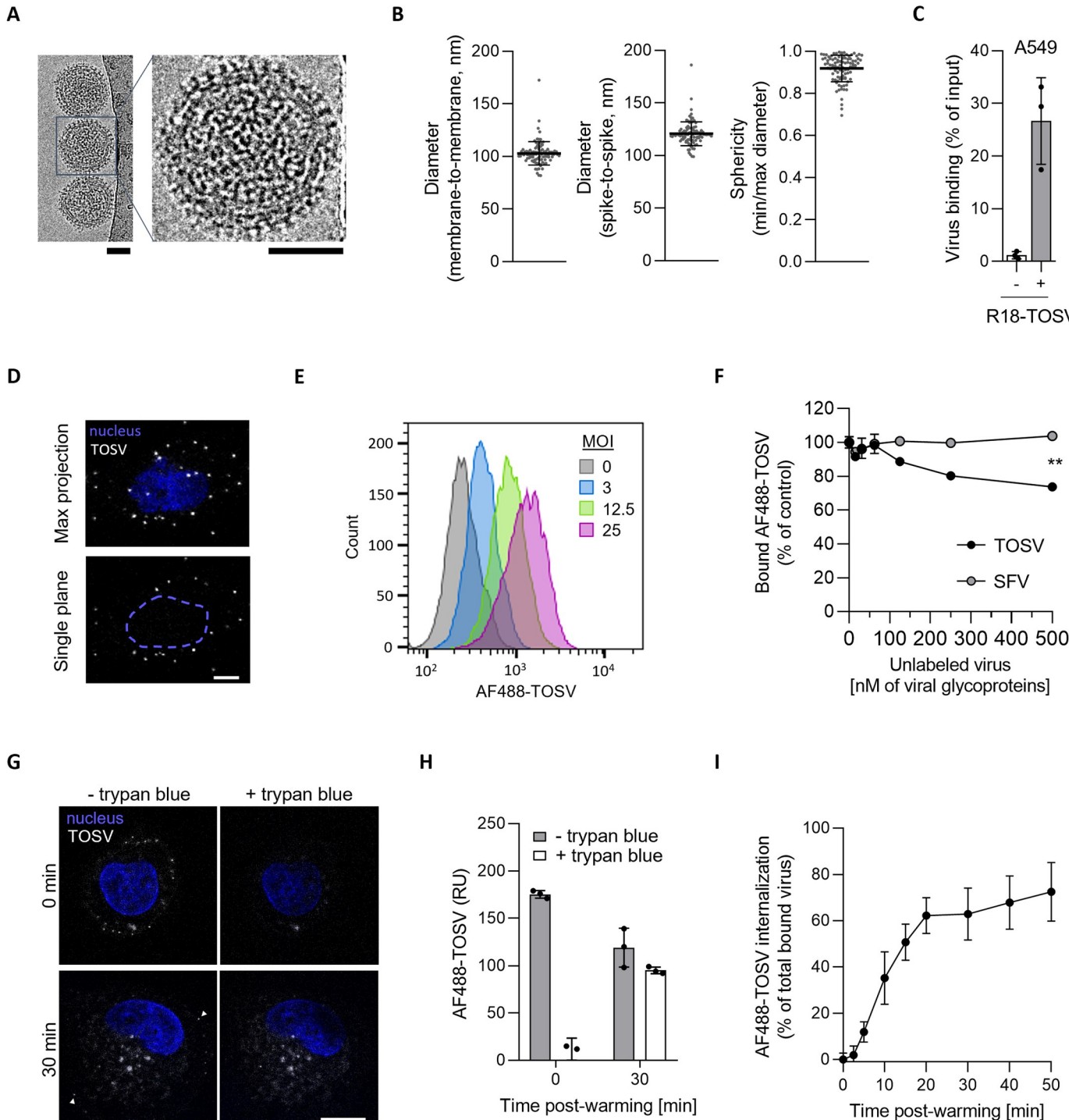

**Fig 2. Toscana virus (TOSV) binds to human A549 cells, and its underlying internalization is slow.** (**A**) TOSV virions were purified and imaged by cryo-electron microscopy (EM). Scale bar, 50 nm. (**B**) The membrane-to-membrane and spike-to-spike diameter of virions was measured from EM images. The sphericity of each TOSV particle was determined as the ratio of the width to the perpendicular length (n = 96). (**C**) R18-labeled TOSV was bound to A549 cells on ice at MOI 10, and samples were analyzed by fluorometry. The virus input was measured in cell-free suspension before binding. (**D**) ATTO647N-labeled TOSV was bound to A549 cells at MOI 1. Nuclei were stained with Hoechst before imaging with a confocal microscope. White spots are cell-associated virions seen in a series of z-stacks merged to one plane (upper panel) and in one focal plane (lower panel). The blue dashed line indicates the nucleus. Scale bar, 5 μm. (**E**) Various amounts of AF488-labeled TOSV (AF488-TOSV) were bound to A549 cells on ice before analysis by flow cytometry. (**F**) A549 cells were pre-incubated with increasing amounts of unlabeled TOSV and Semliki Forest virus (SFV) on ice before the addition of AF488-TOSV (15 nM of glycoproteins) and flow cytometry analysis. Data were normalized to those in cells not pre-exposed to the unlabeled virus. T-test with Welch's correction was applied. **, p<0.01. (**G**) AF488-TOSV

was bound to A549 cells on ice at MOI 10 before warming to 37˚C for 30 min. Samples were treated with trypan blue before confocal imaging. Nuclei were stained with Hoechst. Arrowheads show some fluorescent particles at the cell surface that are quenched upon trypan blue addition. Scale bar, 10 μm. (**H**) Internalization of AF488-TOSV (MOI ~10) was assessed in A549 cells at 30 min with the trypan blue-based assay shown in G, but using flow cytometry for analysis instead of confocal microscopy. RU, relative unit. (**I**) Internalization of AF488-TOSV (MOI ~10) was monitored in A549 cells over 50 min as assayed in H. Internalization is given as the percentage of fluorescence quantified in samples treated with trypan blue compared to that in untreated samples. The fluorescence signal measured in cells not exposed to AF488-TOSV was considered the background signal and subtracted from the other values.

that TOSV binding to cells is specific and likely involves one or more attachment factors or receptors.

## TOSV enters early and late endosomal compartments

The phenuiviruses analyzed for entry have been shown to enter host cells by endocytosis [7]. We, therefore, aimed to determine whether TOSV is internalized and sorted into early endosomes (EEs) following virus binding to the cell surface. To analyze TOSV internalization by fluorescence microscopy, we synchronized virus entry. To this end, we first allowed TOSV-AF488 to bind to A549 cells on ice at high MOI (~10). The cells were washed to remove unbound viruses, then rapidly warmed to 37˚C to trigger endocytosis, and finally placed back on the ice after 30 min to stop further endocytosis. To discriminate between internalized and surface-bound virions, cells were treated with trypan blue for 10 sec before imaging or flow cytometry analysis. Trypan blue is a membrane-impermeable dye that quenches green-emitting dyes such as AF488 and thus quenches the fluorescence emitted by TOSV-AF488 particles exposed on the cell surface while leaving intracellular virions unquenched (Fig 2G and 2H). Time-course analysis of the generation of trypan blue-resistant fluorescence of cell-associated TOSV-AF488 revealed that internalization into A549 cells started within the first 5 min and increased over time to reach the half-maximal level ($t_{1/2}$) within $10 \pm 2$ min and the plateau within 10–15 min later (Fig 2I). Evidently, TOSV uptake occurred rather synchronously.

To assess whether internalization leads to the sorting of viral particles into EEs, we used A549 cells transiently expressing the small GTPase Rab5a, a marker of EEs, tagged with a monomeric enhanced green fluorescent protein (EGFP). After synchronization of TOSV-ATTO647N binding to A549 cells expressing EGFP-Rab5a on ice, the temperature was rapidly shifted to 37˚C for periods of up to 40 min. Confocal microscopy showed TOSV co-localizing with EGFP-Rab5a-positive (+) vesicles 5 min post-warming (Fig 3A). The amount of co-localizing virions reached a maximum within 5–10 min post-warming and decreased thereafter (Fig 3B). In live A549 cells, confocal microscopy showed that TOSV-ATTO647N moved together within EGFP-Rab5a+ endosomal vacuoles (S1 Movie).

In addition, TOSV was observed within vesicles containing the late endosomal markers Rab7a and LAMP-1 tagged with EGFP (EGFP-Rab7a and LAMP-1-EGFP) but at later time points. Co-localization and coordinated motion with EGFP-Rab7a+ endosomes in live A549 cells were maximal 20–40 min after uptake (Fig 3C and 3D and S2 Movie). Co-localization with the lysosome marker LAMP-1 (LAMP-1-EGFP) was somewhat delayed (Fig 3E and 3F). Of note, some virions were located in the middle of the vesicles and had probably not yet undergone fusion with the limiting endosomal membrane. Overall, the results showed that TOSV is sorted into the endocytic machinery after attachment to the cell surface, and the temporal overlap of co-localizations with Rab7 and LAMP-1 suggested that viral particles reach late endosomes (LEs) rather than lysosomes.

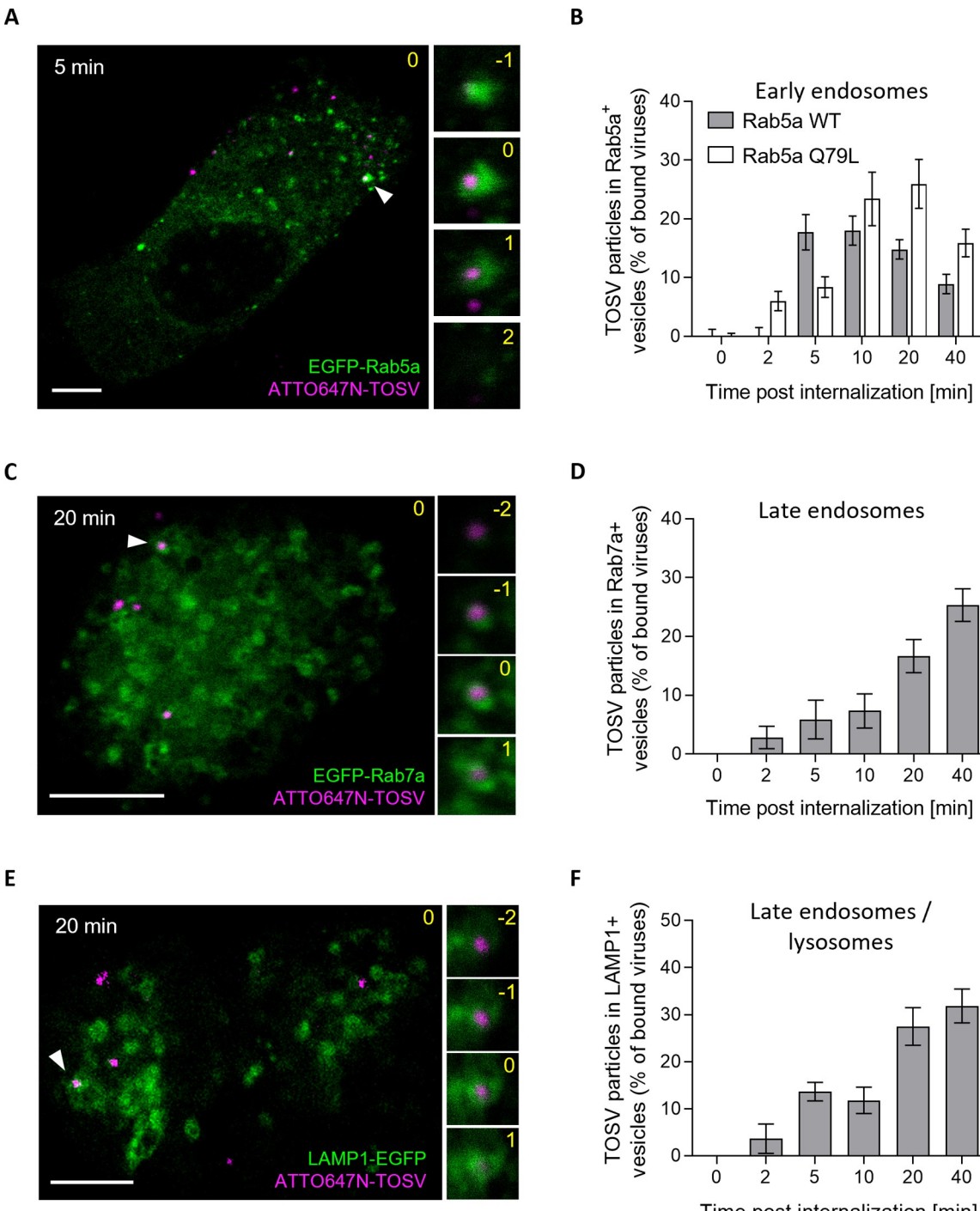

**Fig 3. Toscana virus (TOSV) enters early endosomes and then late endosomal organelles.** (**A**) A549 cells transiently expressing EGFP-Rab5a were exposed to ATTO647N-TOSV at MOI 1 on ice and then rapidly warmed for 5 min at 37°C to allow the internalization of virions. TOSV and Rab5a were imaged by confocal microscopy. One focal plane is shown. Higher magnifications of association between TOSV and Rab5a-positive vesicles (white arrowhead) are shown on the right side as a z-stack series. Yellow numbers indicate the position of the stack in the series, and the original plane is marked with 0. Scale bar, 5 μm. (**B**) Internalization of ATTO647N-TOSV was monitored in cells transiently expressing EGFP-Rab5a wild-type (WT) or its constitutively active mutant Q79L over 40 min as described in A. Co-localization is expressed as the percentage of bound TOSV associated with Rab5a+ vesicles at different times post-warming. A minimum of 6 cells were analyzed per time point. (**C** to **F**) Prebound ATTO647N-TOSV was internalized into A549 cells transiently expressing EGFP-Rab7a (C and D) or LAMP1-EGFP (E and F) for up to 40 min before analysis as described in A and B. At least 9 cells were analyzed per condition. Scale bar, 5 μm.

## TOSV depends on the passage through EEs and LEs for infectivity

To examine whether passage through the endosomal compartments was required for infectivity, we first assessed TOSV internalization and infection in A549 cells transfected with DNA plasmids to express a constitutively-active mutant of Rab5a tagged with EGFP (EGFP-Rab5a Q79L). The expression of this mutant typically results in the enlargement of EEs (Fig 4A), compromising the maturation of LEs and transport of cargo to lysosomes [19]. Infection was measured in populations of cells expressing identical levels of EGFP as selected by flow cytometry. Expression of EGFP-Rab5a Q79L reduced TOSV infection by 80% in comparison to EGFP-Rab5a wild type (wt) (Fig 4B). In contrast to EGFP-Rab5a, the number of virions co-localizing with EGFP-Rab5a Q79L+ vacuoles remained high even after 20–40 min (Fig 3B). Expression of the dominant-negative mutant Rab5a S34N (EGFP-Rab5a S34N), which abrogates the maturation of newly-formed EEs [20], also resulted in a large decrease in infection, *i.e.*, more than 60% (Fig 4B). Altogether, these results indicated that the infectious entry pathway involves the passage of TOSV in Rab5a+ EEs, though the transport to downstream endosomal vesicles is also needed for productive infection.

As a block in the maturation of EEs to LEs impeded infection, TOSV potentially needs to pass through LEs for its productive entry into the cytosol. To test this possibility, we evaluate the role of the small GTPase Rab7a in TOSV infection. Rab7a is a key player in LE maturation and functions [21]. When A549 cells transiently expressed an EGFP-tagged dominant negative mutant of Rab7a (EGFP-Rab7a T22N), TOSV infection was severely impaired, but not when the cells expressed the constitutively active Q67L mutant of Rab7a (Fig 4C). In some cell types, LE maturation relies on microtubule (MT)-mediated transport of endosomes to the perinuclear region of cells and on free cytosolic ubiquitin [22, 23]. Treatment of neurons and A549 cells with either nocodazole or colcemid, two drugs that hamper MT polymerization, resulted in a 40–60% decrease in infection (Fig 4D and 4E). When free ubiquitin was depleted by the proteasome inhibitor MG-132, TOSV infection was reduced in a dose-dependent manner in both neurons and A549 cells. Conversely, taxol, an MT-stabilizing drug, did not affect the infection of A549 cells. Taken together, these experiments show that the transport of virions to LEs is required for infectivity.

## Low pH is sufficient and necessary for TOSV fusion

To examine whether the acidic pH in endosomal vesicles is important in TOSV infection, the virus was added to neurons and A549 cells in the presence of agents that neutralize vacuolar pH. The lysosomotropic weak bases ammonium chloride ($NH_4Cl$) and chloroquine induced a dose-dependent inhibition of TOSV infection (Fig 5A–5C). In these experiments, we monitored and regulated the medium for a pH above 7.0 ensuring the effectiveness of the base. Two inhibitors of vacuolar-type H+-ATPases, bafilomycin A1 and concanamycin B, gave similar results. Taken together, these experiments showed that TOSV depends on vacuolar acidification for infection in both neurons and A549 cells. These results also suggested that the virus penetrates host cells by acid-activated membrane fusion.

To define the pH threshold and link fusion with infection, we tested the capacity of TOSV to fuse at the plasma membrane of cells as originally described for SFV [24]. In such a scenario, the virus bypasses the need for endocytosis during productive infection. Briefly, TOSV was bound to A549 cells at MOI 10 at 4°C, the temperature was rapidly elevated to 37°C for 1.5 min in buffers with different pH values, and $NH_4Cl$-containing medium at neutral pH was then added for the remaining period of infection to prevent further infection through endosomes. The bypass resulted in efficient infection at pH values of 5.7 and below (Fig 6A). 50% of the maximal infection was reached at a pH of 5.6. These data demonstrated that a reduction in

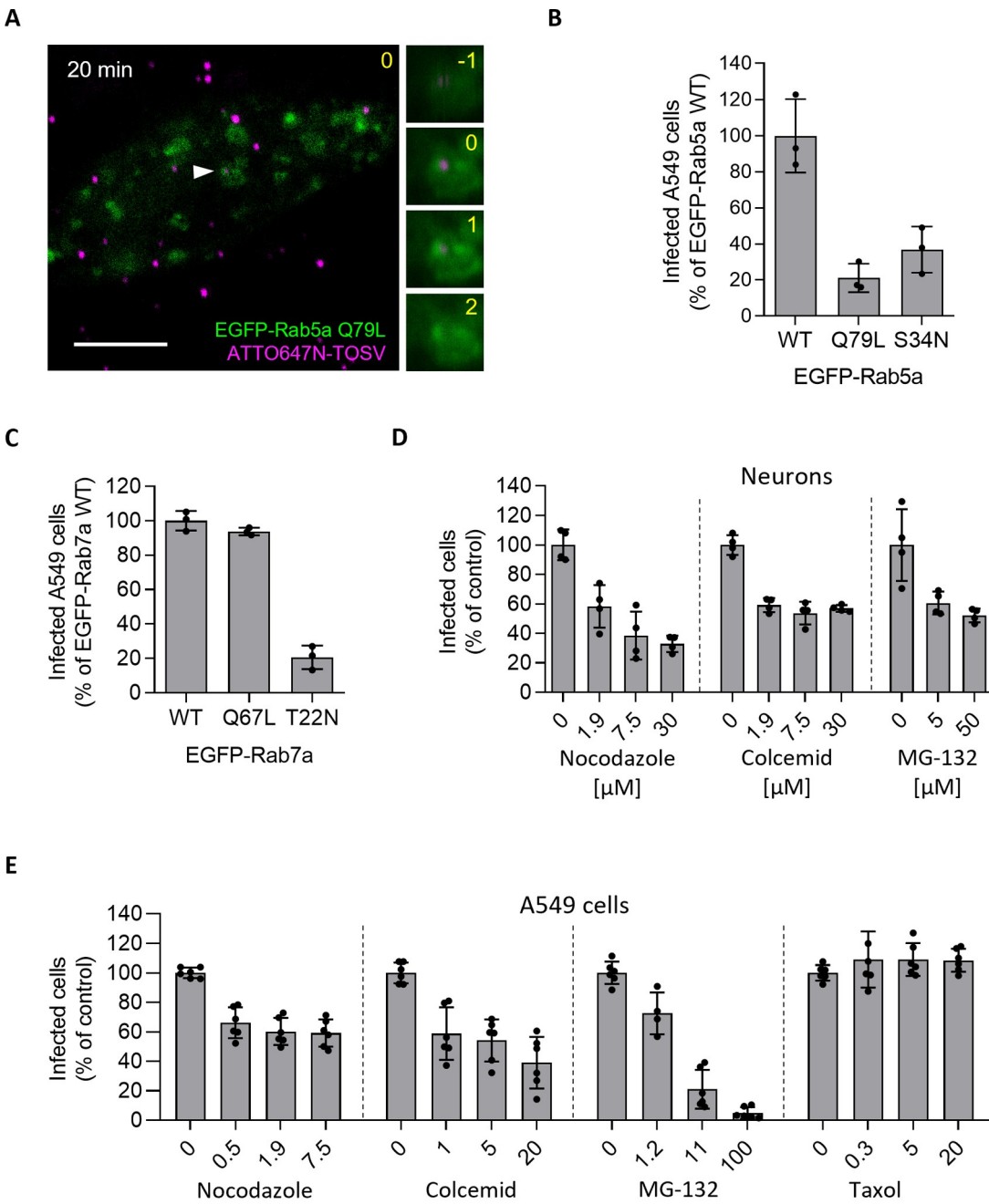

**Fig 4. Toscana virus (TOSV) relies on late endosomal maturation for infectious entry.** (**A**) ATTO647N-TOSV (MOI ∼1) was internalized for 20 min in A549 cells transiently expressing the constitutively active mutant Q79L of EGFP-Rab5a and then imaged by confocal microscopy as described in Fig 3. One focal plane is shown. Higher magnifications of association between TOSV and Rab5a Q79L-positive vesicles (white arrowhead) are shown on the right side as a z-stack series. Yellow numbers indicate the position of the stack in the series, and the original plane is marked with 0. Scale bar, 5 μm. (**B**) A549 cells transfected to transiently express EGFP-Rab5 WT, its constitutively active mutant Q79L, and its dominant-negative mutant S34N were challenged with TOSV at MOI 4 for 6 h. Using flow cytometry, cell populations with similar levels of Rab5a were selected and analyzed for infection. Infection was normalized to that in the cell population expressing EGFP-Rab5 WT. (**C**) The mutants Q67L (constitutively active) and T22N (dominant-negative) of Rab7a were assessed for their effect on TOSV infection in A549 cells as described in B. (**D** and **E**) iPSC-derived neurons (D) and A549 cells (E) were pretreated with nocodazole, colcemid, MG-132, or taxol and then infected with TOSV at MOI 10 and 2, respectively, in the continuous presence of inhibitors. Infection was quantified by flow cytometry, and data were normalized to those in control samples without inhibitor treatment.

**A**

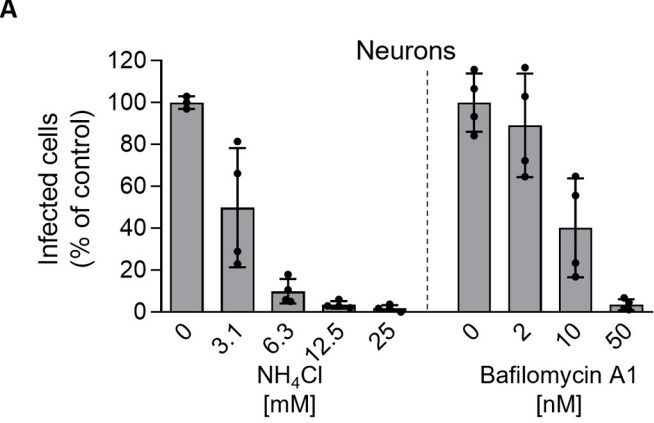

**B**

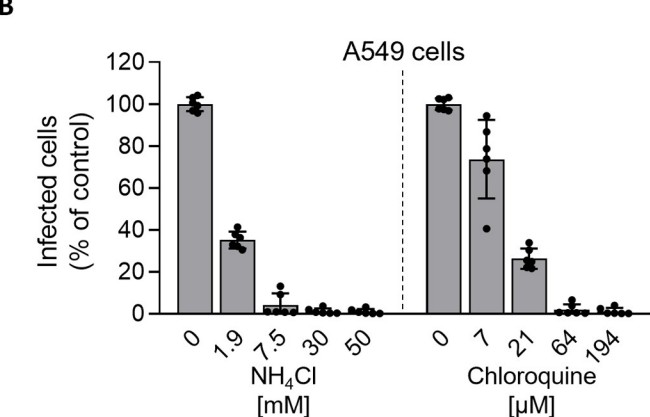

**C**

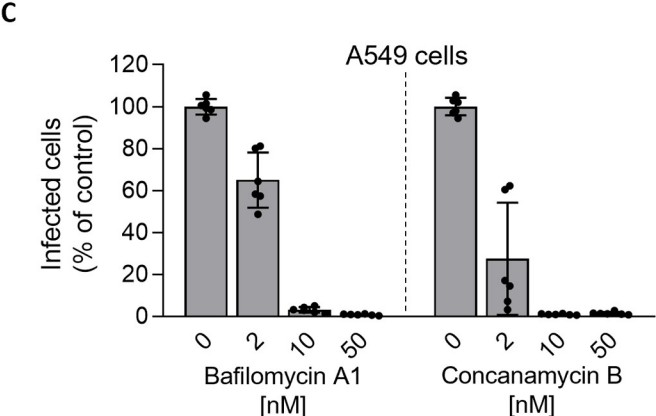

**Fig 5. Toscana virus (TOSV) entry into iPSC-derived human neurons depends on endosomal acidification.** (**A** to **C**) iPSC-derived neurons (A) and A549 cells (B and C) were pretreated with agents that elevate endosomal pH at the indicated concentrations and were infected with TOSV at MOI 10 for 8 h and MOI 2 for 6 h, respectively, in the continuous presence of ammonium chloride (NH$_4$Cl), chloroquine, bafilomycin A1, and concanamycin B. Infection was analyzed by flow cytometry, and data were normalized to those of control samples without inhibitor treatment.

pH is sufficient to trigger infectious penetration of viral RNPs from the plasma membrane to the cytosol. Additional processing steps were apparently not required in the endosomes to activate fusion or infection by TOSV.

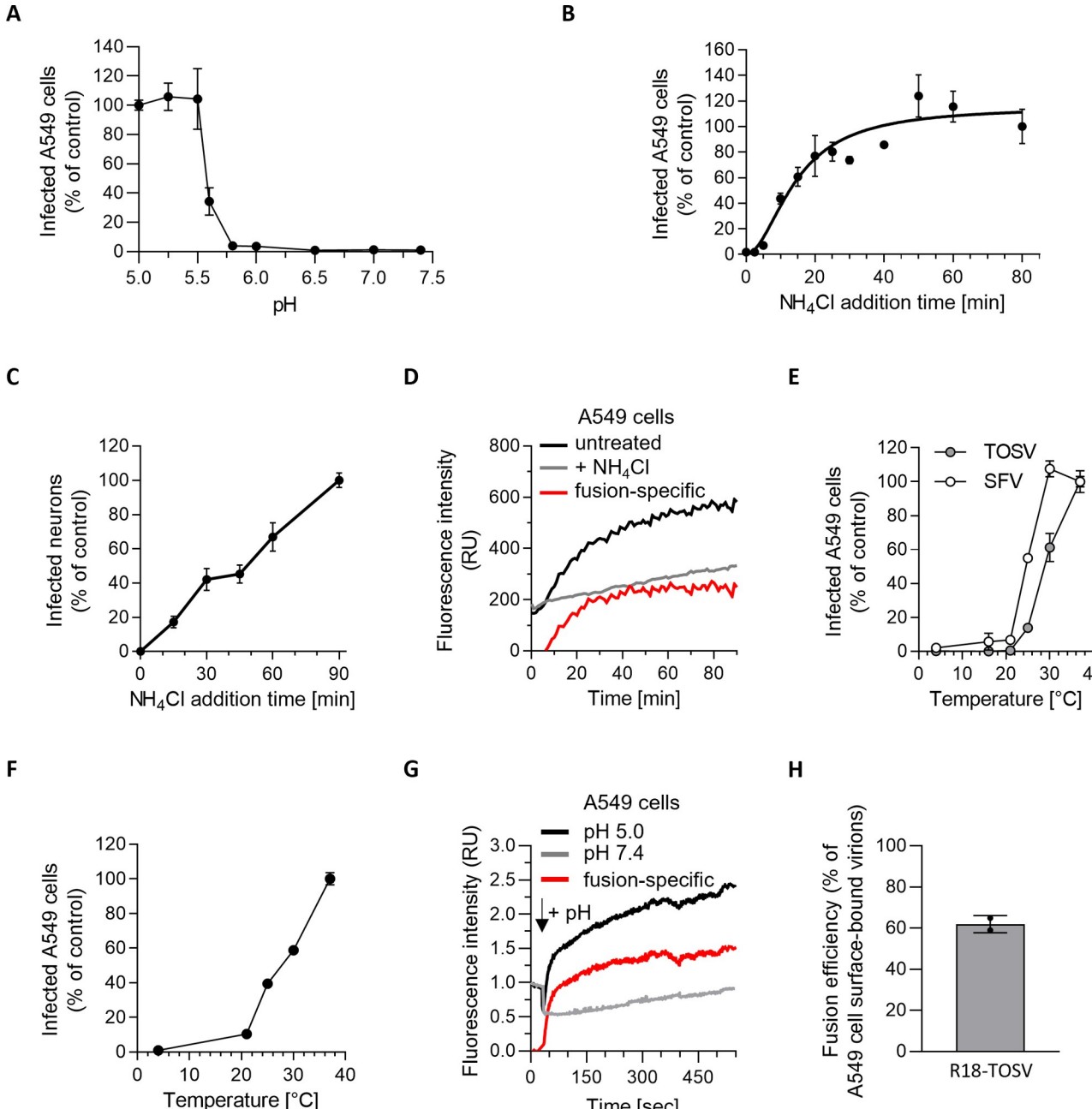

**Fig 6. Toscana virus (TOSV) penetrates host cells by acid-activated membrane fusion.** (**A**) A549 cells were exposed to TOSV (MOI ∼10) on ice, subjected to various pH values at 37°C for 90 sec to trigger the virus fusion at the plasma membrane, and then incubated for 7 h at 37°C in the presence of 50 mM NH$_4$Cl to prevent viral penetration from endosomes. Infection was quantified by flow cytometry, and the data were normalized to those from samples where the infection was triggered with a buffer at pH 5.0. (**B** and **C**) TOSV particles were bound to A549 cells (B) and iPSC-derived neurons (C) at MOIs 1 and 15, respectively, on ice and then rapidly shifted to 37°C to allow virus internalization. 50 mM NH$_4$Cl was added at the indicated times to block further viral penetration. Infected cells were quantified by flow cytometry, and data were normalized to the samples where NH$_4$Cl was added 80 min (B) and 90 min (C) post-warming. (**D**) R18-TOSV was bound at MOI 10 to A549 cells on ice and rapidly warmed to 37°C. The increase in fluorescence resulted from the dequenching of the lipid dye R18 after virus fusion with cell membranes, as measured by fluorometry (black line). NH$_4$Cl was used to block virus fusion by neutralizing endosomal pH and, thus, to define the fluorescence background due to spontaneous translocation of the R18 dyes between the viral envelope and the neighboring cell membrane (grey line). The red line shows the virus fusion-specific R18 release, *i.e.*, the black line (fusion + free diffusion) minus the grey line (free diffusion). RU, relative unit. (**E**) TOSV and Semliki Forest virus (SFV) were bound to A549 cells on ice, and samples were shifted to indicated temperatures for 50 min. Infected cells were then incubated at 37°C for 6 h in the presence of NH$_4$Cl and quantified by flow cytometry. Infection was normalized to that in samples incubated throughout at 37°C. (**F**) TOSV fusion was assessed in A549 cells for efficiency at various temperatures using the assay in A. Data were normalized to those of samples incubated throughout at 37°C. (**G**) The

binding of R18-TOSV (MOI $\sim$10) to A549 cells was synchronized on ice. Samples were then warmed to 37˚C in a fluorometer, and the fluorescence signal was monitored over 10 min. "+pH" indicates when buffers at pH 5.0 or 7.4 were added to trigger virus fusion. Data were normalized to those at the time point 0. The red line shows the virus fusion-specific R18 release at pH $\sim$5.0. RU, relative unit. (**H**) R18-TOSV penetration into A549 cells at MOI 10 was recorded in real time for 90 min using the protocol in D, and triton X-100 was added at the end to induce the dequenching of all R18 molecules associated with bound and internalized virions. The data show the ratio between the fluorescence resulting from viral fusion and that associated with all virions in the cells.

## Acid-activated penetration occurs in late endosomal compartments

To determine the timing of the acid-requiring step following virus internalization, we took advantage of the fact that the rise in endosomal pH is almost instantaneous when $NH_4Cl$ is added to the extracellular medium [25]. To synchronize virus entry, virions were first allowed to bind to neurons and A549 cells on ice at MOIs of 15 and 1, respectively. Virus entry was then allowed by rapid warming to 37˚C, and $NH_4Cl$ was added at different times following the temperature switch. $NH_4Cl$ was used at 50 mM to ensure that infection was completely abolished after adding the weak base. Infectious penetration started after a 5-min delay in A549 cells and reached a $t_{1/2}$ within 15 min and a plateau 25–45 min later (Fig 6B). It was apparent that individual viral particles had completed the $NH_4Cl$-sensitive step non-synchronously in neurons, most likely due to the heterogeneity of cell preparations typical of iPSC-derived neurons (Fig 6C).

To further analyze acid-activated membrane fusion in late endosomal vacuoles, we relied on TOSV-R18 to monitor viral fusion in living cells with a fluorimeter. In this assay, the increase in fluorescence signal results from dequenching of the fluorescence lipid dye R18 upon activation of viral fusion. Though the subsequent release of the dye into the target cell membranes corresponds to the hemifusion mixture of outer leaflets and not to fusion pore formation, it is a good correlate for fusion. TOSV-R18 was bound to A549 cells on ice, and virus endocytosis was synchronized by switching the cells rapidly to 37˚C (Fig 6D). The kinetics measured in this assay were very similar to those determined with the above procedure based on $NH_4Cl$ addition. The fluorescence signal started to increase after a 6 min lag and reached a $t_{1/2}$ within 18 min ± 2 min post-warming with a plateau value about 20 min later.

The kinetics of penetration closely resembled the time course of endolysosomal maturation, which usually lasts 30–60 min [22]. To challenge the hypothesis that TOSV penetration occurs in LEs, we examined the temperature dependence of entry. The transport of cargo from EEs to LEs is known to be inhibited at temperatures below 20˚C [26]. TOSV binding to A549 cells was synchronized on ice, and cells rapidly warmed to different temperatures for 1 h before incubation at 37˚C for 6 h in the presence of $NH_4Cl$ to prevent further penetration. The infection was greatly reduced at 30˚C and below (Fig 6E). In contrast to TOSV, a temperature of 30˚C had no noticeable effect on infection by SFV, which is acid-activated in EEs [27]. SFV infection was still detected at temperatures as low as 16˚C.

To rule out that the fusion process was altered by temperature, we also analyzed the temperature dependence of TOSV fusion using the bypass protocol described above. At 25˚C, fusion corresponded to 40% of the 37˚C control (Fig 6F), whereas infection via the normal route was lowered by 85% (Fig 6E). At 21˚C, fusion was still 10% of the 37˚C control, whereas infection via the normal route could no longer be detected. As expected of a virus capable of replicating in insect hosts, this suggested that fusion was not a bottleneck for penetration at lower temperatures. Most likely the viral particles did not infect at 25˚C and below because they did not reach LEs with low enough pH.

Next, we analyzed the dynamics of viral fusion. To this end, we induced fusion of TOSV-R18 at the plasma membrane of cells, as described in the bypass assay, but instead measured the increase in fluorescence associated with the dequenching of R18 dye in real time. In

brief, R18-labeled virions were first allowed to attach to A549 cells on ice, and viral fusion was then triggered by the addition of buffer at pH $\sim 5.0$. We found that the release of R18 molecules into target cell membranes reached a $t_{1/2}$ of 27 ± 16 sec and was completed 25–50 sec later (Fig 6G). Almost two-thirds of the plasma membrane-bound virions entered the cells and fused (Fig 6H). Together, our data indicate that acid-activated penetration of TOSV involves most cell surface-bound virions and is achieved through a fast and efficient fusion process from late endosomal compartments.

## The TOSV envelope glycoprotein Gc is a class-II fusion protein

A growing body of evidence suggests that the Gc envelope glycoprotein of phenuiviruses belongs to the group of class-II membrane fusion proteins [7]. As a result, phenuiviruses are thought to follow an acid-activated penetration process in line with this class of fusion proteins [28, 29]. No structural data are currently available for TOSV glycoproteins, but others have resolved the crystal structure of RVFV Gc [10]. Analysis of the M polypeptide sequence with the blastp algorithm showed that TOSV Gc shares about 48% amino acid identity and 67% similarity with RVFV Gc (Figs 7A and S3). The fusion domain of RVFV reaches 83% amino acid identity with the corresponding sequence in TOSV Gc. This high degree of conservation suggests a fairly recent evolutionary divergence between TOSV and RVFV.

The level of identity and similarity in amino acids supports the idea that the Gc glycoproteins of TOSV and RVFV resemble each other structurally. To further test this possibility, we utilized AlphaFold to predict the structure of the TOSV Gc ectodomain based on the information available for the RVFV Gc structure (S1 File). A bent conformation was obtained by comparison with intact RVFV virions (Protein Data Bank [PDB], 6F9F) [16], likely corresponding to the canonical pre-fusion orientation of the glycoprotein (Fig 7B). The Gc AlphaFold prediction exhibited domains I, II, and III typical of its phenuivirus orthologs and other viral class-II fusion proteins. The acid-activated conformation of TOSV Gc (S2 File) was predicted using the X-ray post-fusion structure of RVFV Gc (PDB, 6EGT) [30] (Fig 7C). The extent of confidence in the modeling was assessed using an average predicted local distance difference test (pLDDT). The closer the value is to 100, the more likely the prediction is to be close to the real structure. A value below 50 is categorized as low confidence. Both structural models of TOSV Gc achieved a high overall confidence score, with pLDDT values higher than 90 for the individual domains I, II, and III.

The AlphaFold pre-fusion conformation of TOSV Gc was virtually identical to the experimentally-resolved pre-fusion structure for RVFV Gc, except for a 19-degree greater angle between domains I and II (Fig 7D). This difference could be due to the preference of Alpha-Fold for more energetically stable conformations and not to the original pre-fusion Gc structure itself. The post-fusion Gc forms of the two viruses differed further. The position of $A_0B_0$ and $C_0D_0$ strands in domain I did not undergo major changes after TOSV Gc activation (Fig 7E and S3 Movie). Both $A_0B_0$ and $C_0D_0$ strands had deviations of 101 and 34 degrees, respectively, from the same strands in the RVFV Gc post-fusion structure. A second distinction was the barrel sheet in domain III, which was lower in position for TOSV than for RVFV, and the third difference was the presence of an isoleucine residue (I967) in the fusion loop instead of a tryptophan (W821). This latter difference is consistent with the observation made by others that isoleucine at this position allows the distinction between phenuiviruses transmitted by sand flies and ticks and other phenuiviruses [30]. Overall, our computational approaches showed that TOSV Gc can be confidently considered a viral class-II fusion protein.

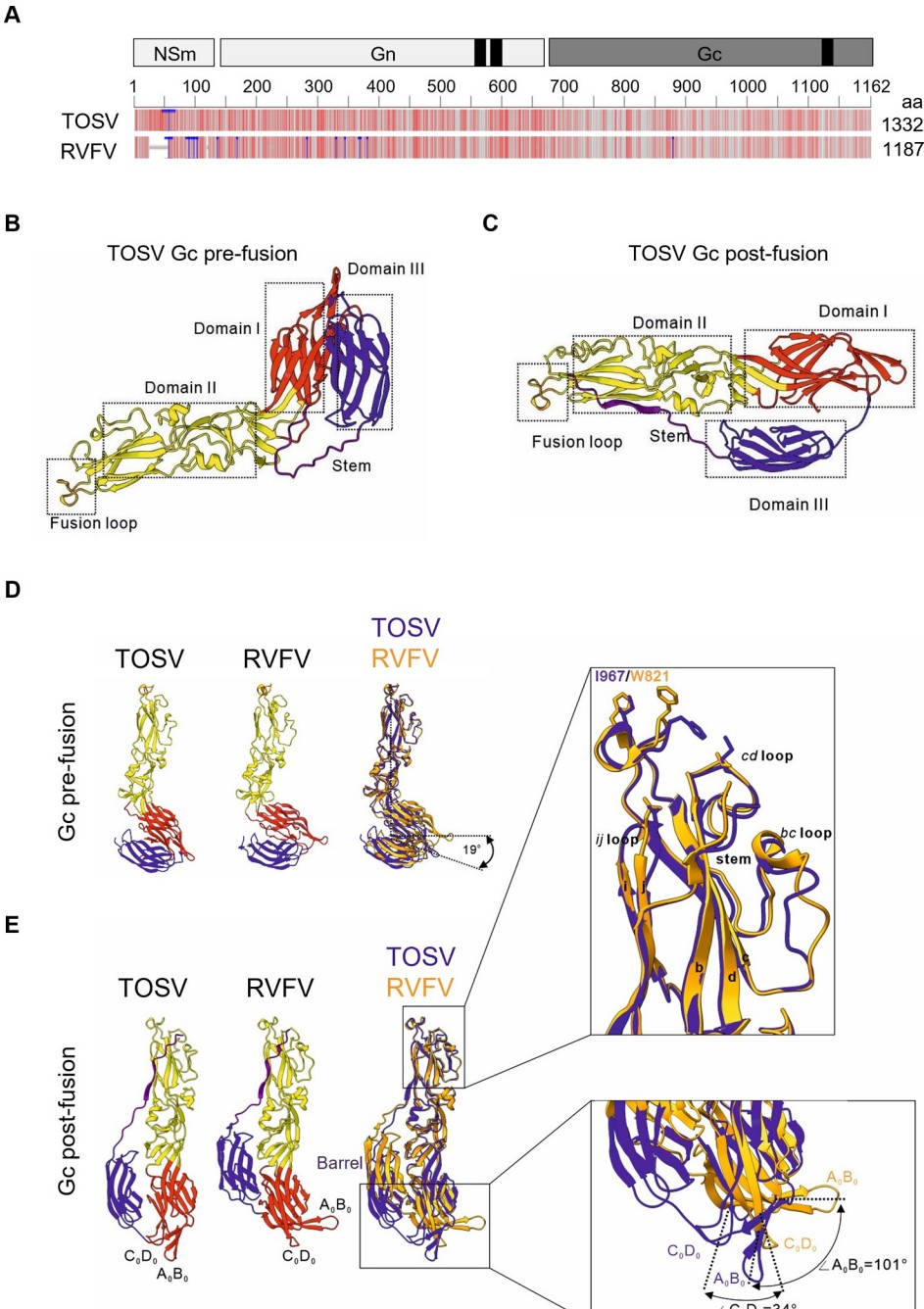

**Fig 7. Toscana virus (TOSV) Gc belongs to the group of class-II fusion proteins.** (**A**) The M polyproteins of TOSV (strain H4906) and Rift Valley fever virus (RVFV) (strain 35/74) were aligned with blastp suite-2sequences and Multiple Sequence Alignment Viewer. The black boxes indicate the transmembrane domains and the blue lines indicate the insertions into TOSV and RVFV M polyproteins. The grey lines correspond to identical amino acids, the light red lines to similar amino acids, and the dark red lines to different amino acids. (**B** and **C**) Pre- (B) and post-fusion (C) conformations of TOSV Gc were predicted using the AlphaFold2-derived algorithm ColabFold, and the Gc structures available for RVFV as modeling models (PDB, 6F9F and 6EGT). The structural predictions were visualized using PyMOL. Domains I, II, and III are typical of class-II membrane fusion proteins with the fusion loop at the end of domain II and appear in red, yellow, and blue, respectively. Flexible stem-loops are shown in purple. (**D** and **E**) Pre- (D) and post-fusion (E) conformations of TOSV Gc were compared with those of RVFV Gc using UCSF ChimeraX. In the Gc structure overlay, TOSV and RVFV Gc appear in blue and orange, respectively. Otherwise, the color code for domains I, II, and III is the same as that used in B and C. The right boxes in E show magnifications of the fusion unit with corresponding loops and amino acids (upper panel) and the position of $A_0B_0$ and $C_0D_0$ strands (lower panel).

## TOSV remains infectious even after acid exposure below the fusion threshold

The activation and priming of class-II viral fusion proteins are described as irreversible steps, and viral fusion proteins act only once [29]. To check whether TOSV fusion is an irreversible process, we first evaluated the possibility of inactivating the virus, by applying acidic buffers in the absence of target-cell membranes, before infection under neutral-pH conditions. In such an assay, the virus undergoes a transition toward the post-fusion state at the optimal pH. If the transition is irreversible, the fusion protein is no longer able to fuse with target-cell membranes, and, thus, the viral particles are rendered noninfectious. Using this approach, we did not observe any negative effect on TOSV infectivity in A549 cells by exposing virions to buffers ranging from pH $\sim$ 5.0 to 7.5 for 5 min (Fig 8A). Infection was even greater when the viral particles were subjected to buffers at pH $\sim$ 6.0 and 6.5.

To determine whether low pH resistance is a hallmark of TOSV or has larger implications, we expanded our study to related and unrelated viruses with class-I or class-II fusion proteins. Briefly, RVFV, Germiston virus (GERV), and UUKV are bunyaviruses with a class-II fusion protein and related to TOSV. In addition, SFV is an alphavirus with a class-II membrane fusion protein, and the unrelated influenza A virus (IAV) has a class-I fusion protein [29]. SFV enters host cells from EEs with a fusion threshold at pH $\sim$ 6.2 while the others are late-penetrating viruses (L-PV) with a pH-activation threshold ranging from 5.0 to 6.0 depending on the virus species [19, 31–34]. Like TOSV, we found that the three other bunyaviruses remained infectious in A549 and BHK-21 cells after exposure to buffers of different pH values (Fig 8B–8D). In contrast to bunyaviruses, the infectivity of SFV and IAV was dampened by 90–95% after exposure to pH below 6.0 (Fig 8D and 8E). We noted that although TOSV infectivity appeared to be lower in BHK-21 cells after exposure to pH $\sim$ 5.5, 20–30% of virions remained infectious, whereas SFV inactivation was virtually complete (Fig 8D).

To investigate whether the fusion process as such was affected by greater virus binding to cells following exposure to low pH, ATTO647N-TOSV was first subjected to buffers ranging in pH from 5.0 to 7.4. The fluorescently-labeled virions were then allowed to attach to A549, BHK-21, and Vero cells under neutral pH conditions on ice for 1.5 h before imaging (Fig 8F and 8G). Lowering the pH to 6.0 in the binding medium resulted in a higher efficiency of virion attachment to most cell lines, from $\sim$ 10 to $\sim$ 20 virions per cell. Similar results were obtained when UUKV binding to BHK-21 cells was analyzed by flow cytometry (Fig 8H). More acidic pH probably caused the unmasking of epitopes in viral glycoproteins that promote interactions with one or more cell attachment factors or receptors. The number of cell-bound virions decreased slightly at the lowest pH, *i.e.*, 5–10 virions per cell at pH $\sim$ 5.0. Overall, virus binding could not account for enhanced fusion at pH $\sim$ 5.5 and below. Combined, the results indicated that TOSV is not inactivated at pH values below the fusion threshold. This suggested that virions remain infectious in late endosomal vacuoles even when the intraluminal acidity was inferior to the fusion threshold.

## A mildly acidic environment primes low-pH-activation of TOSV fusion in LEs

To further examine the effect of pH on TOSV activation and fusion, we induced TOSV-R18 fusion on the A549 cell surface with buffers ranging in pH from 5.0 to 5.8 as described above and monitored the dye dequenching in real time. Viral fusion could be detected for pH as high as 5.8 and the lower the activation pH, the faster the fusion dynamics (Fig 9A). In the pH range of 5.0 to 5.5, half of the bound virions had completed fusion within 27–57 sec (Fig 9B). At pH of 5.8, the $t_{1/2}$ was reached within 164 sec, *i.e.*, the process took longer than at lower pH

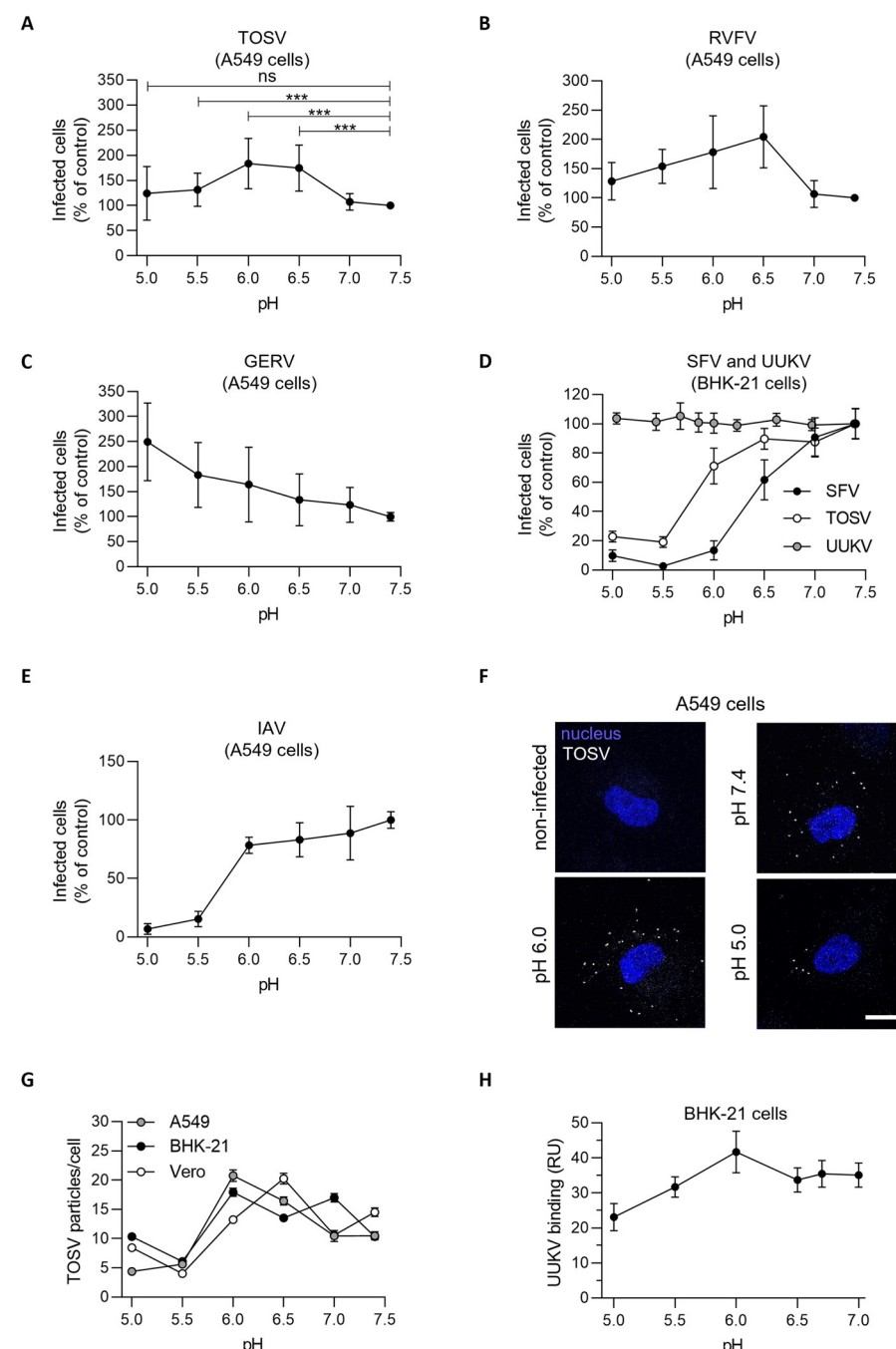

**Fig 8. Low pH does not inactivate Toscana virus (TOSV) or other bunyaviruses.** (**A**) TOSV (MOI ∼2) was first pretreated at various pH for 5 min and then buffered at pH 7.4 before the infection of A549 cells and quantification by flow cytometry. Data were normalized to that of samples pretreated at pH 7.4. T-test with Welch's correction was applied. ***, p<0.001; ns, nonsignificant. (**B** to **E**) The sensitivity of Rift Valley fever virus (RVFV) construct RVFVΔNSs:EGFP, Germiston virus (GERV), Semliki Forest virus (SFV), Uukuniemi virus (UUKV), TOSV, and influenza A virus (IAV) to acidic pretreatments was evaluated as described in A. (**F**) ATTO647N-TOSV at MOI 1 was pretreated at low pH and then allowed to bind to A549 cells on ice before confocal imaging. Nuclei were stained by Hoechst. Scale bar, 10 μm. (**G**) Depicted is the quantification of viral particles bound to A549, BHK-21, and Vero cells as described in F. n > 124 cells. (**H**) Alexa Fluor 647-labeled UUKV was pretreated at low pH and bound to BHK-21 cells at MOI 0.3 as described in F. Virus binding was quantified by flow cytometry, and data were normalized to samples pretreated at pH 7.0.

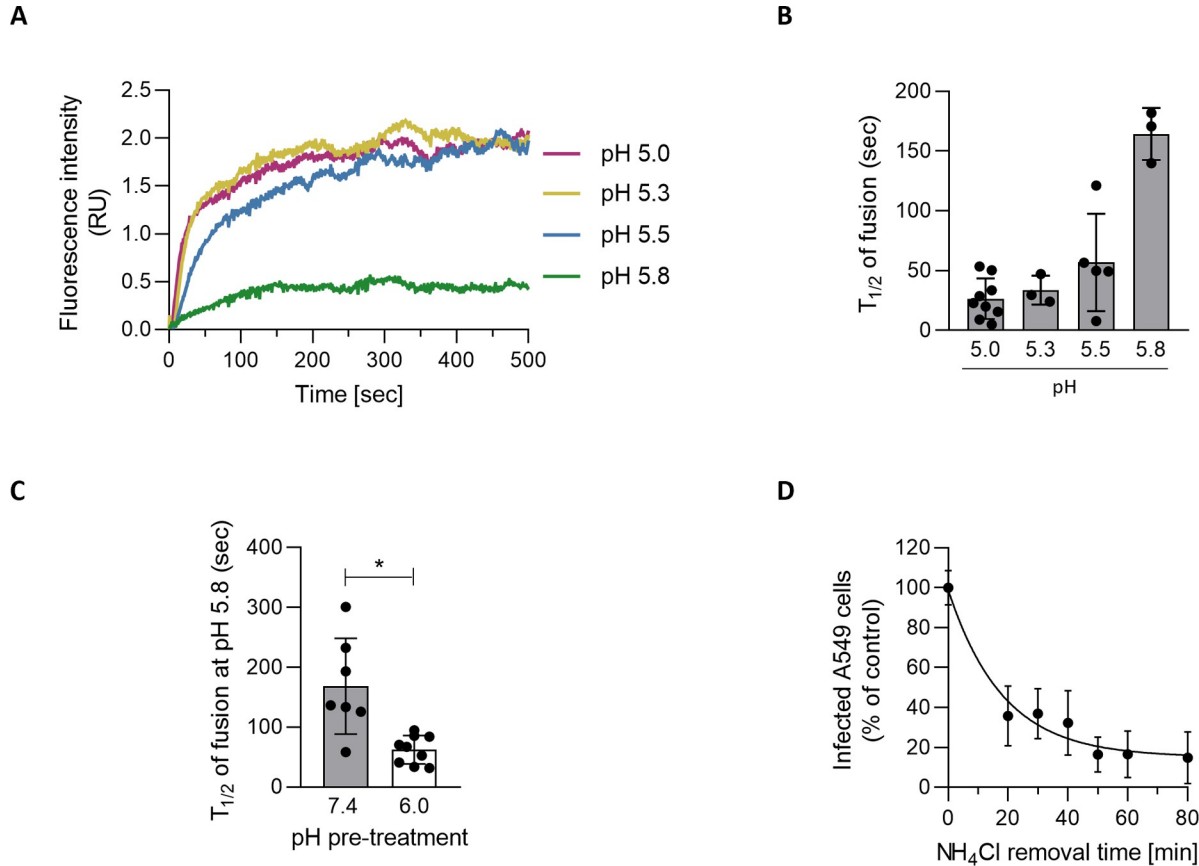

**Fig 9. Toscana virus (TOSV) shows remarkable adaptability to the endosomal environment to penetrate cells.** (**A**) The binding of R18-TOSV (MOI ~10) to A549 cells was synchronized on ice. Cells were then rapidly warmed to 37˚C, and virus fusion was triggered by adding acidic buffers within a fluorometer while recording the fluorescence signal. The virus fusion-specific R18 release is shown. Data were normalized to those at the time point 0. RU, relative unit. (**B**) The half-maximal fluorescence intensity ($t_{1/2}$) was measured in the series of data obtained in A. n > 3. (**C**) R18-TOSV particles were pretreated at pH 6.0 or 7.4 and then neutralized as described in Fig 8A before being assessed and analyzed as in panel A. The $t_{1/2}$ of fusion was calculated from n > 7. T-test with Welch's correction was applied. *, p<0.05. (**D**) After the synchronization of TOSV binding at MOI 1 on ice, A549 cells were rapidly warmed to 37˚C in the presence of $NH_4Cl$ (50 mM). $NH_4Cl$ was then washed out at the indicated times to allow endosomal acidification and the acid-activated penetration of infectious TOSV particles. Samples were harvested 6 h later, and infection was analyzed by flow cytometry. Values were normalized to those from samples for which $NH_4Cl$ was removed at t0.

values. When TOSV was first pre-exposed to mildly acidic pH values such as those prevalent in EEs, we found that virions were then able to fuse markedly faster at a pH of 5.8 (Fig 9C), a pH value typically found in LEs at the beginning of their maturation. The $t_{1/2}$ of the fusion decreased from 168 to 62 sec, which is somewhat similar to the result obtained with an activation pH of 5.5 without pretreatment. Overall, the data demonstrated that TOSV can achieve fusion at higher pHs when exposed to mildly acidic pH for longer periods. The passage through EEs and exposure to a mildly acidic environment most likely favor the activation of TOSV fusion at lower pHs in LEs.

## TOSV remains acid-activable in endosomal vesicles for long periods

The resistance of TOSV to low acidity and the ability of virions to fuse at various pH values led us to postulate that the virus is less prone to inactivation in the endocytic machinery. To examine how long TOSV remains acid-activable in endosomal vesicles, we reversed the approach of

adding NH$_4$Cl. Virus binding to A549 cells was synchronized on ice, and cells were rapidly warmed in the presence of NH$_4$Cl before the weak base was washed out at different times. This assay relies on the fact that the neutralization of endosomal pH by NH$_4$Cl is reversible after washing. In other words, we determined the time at which TOSV acid activation was no longer possible. In A549 cells, infection decreased by 60% during the first 30 min and then more slowly until it reached an 80% decrease after 80 min (Fig 9D). Altogether, these experiments indicated that TOSV infectivity remains high in endosomal vesicles if the virus is not allowed to enter the cytosol after a long period.

Together, our results showed that TOSV resembles late-penetrating viruses in that its entry depends on a normal maturation of LEs. It is transported from EEs to LEs, and its pH of fusion corresponds to that prevailing in LEs. It cannot infect cells in which cargo transport into the degradative branch of the endocytic pathway is blocked. TOSV, and possibly other bunyaviruses, differs remarkably from other acid-activated viruses in its use of endosomal pH as a cue to enter host cells.

## Discussion

TOSV is a re-emerging human pathogen in southern Europe and northern Africa with more than 250 million people potentially exposed and up to 50% seroprevalence in some areas of the Mediterranean basin [4]. The TOSV life cycle is, however, poorly characterized, and overall, this virus remains neglected. Here, we developed reliable and accurate assays to study the early steps of TOSV infection in human cell lines and human iPSCs-derived brain cells, which are targeted in the late stages of infection [35]. We applied flow cytometry, fluorescence microscopy, and fluorimetry to analyze each stage of the TOSV entry program, from virus binding and uptake to intracellular trafficking and membrane fusion. To track single viral particles, we labeled TOSV with fluorescent amine-reactive dyes and took advantage of the free amine residues in the viral glycoproteins Gn and Gc. In addition, we relied on the autoquenching property of the lipid dye R18 at high concentrations to examine the acid-activated membrane fusion of virions.

The binding of virions to the cell surface was specific, although a minority of the total input virus was attached to cells. One reason is likely related to the biochemical and biophysical properties of TOSV receptors, the identity of which remains largely undetermined. Similar observations have been made for other phenuiviruses, such as UUKV [19]. Confocal microscopy images showed that the number of bound virions per cell was significantly higher than the MOI. This result indicated that a low fraction of viral particles in the virus stocks were infectious, with only one virion every 10. Still, this ratio was high compared to that of other phenuiviruses, *e.g.*, the ratio in UUKV infectivity is lower than 1:1,000 [19]. Altogether our results are consistent with a recent study showing that two distinct incomplete phenuivirus populations, which are unable to spread autonomously due to the lack of one or more genome segments, can cooperatively support infection and spread [36].

The penetration of enveloped viruses relies on the fusion between the virion envelope and cell membranes. In most cases, virions are endocytosed after binding, and fusion is triggered in endosomes after the acid activation of viral glycoproteins. The first observation that TOSV follows this penetration strategy was the sensitivity of infection to agents that elevate endosomal pH, as is typical for other phenuiviruses [19, 31, 37]. Using NH$_4$Cl, we showed that the first incoming, infectious particles reached the acid-dependent step 5 min after cell warming and that half had completed this step within 15 min. The timings resembled those of viruses penetrating from LEs such as UUKV and IAV [38]. These viruses pass the acid-sensitive step typically with a t$_{1/2}$ of 15–20 min. For comparison, viruses fusing in EEs, such as SFV or

vesicular stomatitis virus, become $NH_4Cl$ insensitive within 3–5 min as their internalization is almost instantaneously followed by acid activation [39].

The fact that TOSV has indeed an acid-triggered membrane fusion activity was demonstrated with the fusion of cell-bound viruses to the plasma membrane. In that case, fusion was triggered at a pH of 5.6 and below, an acidity typical of late endosomal compartments. Cryo-electron microscopy images of TOSV showed roughly round particles, homogeneous in size with an average diameter of 103 nm and spike-like projections of 9 nm, quite similar to other phenuiviruses such as UUKV and RVFV [8, 9]. Strikingly, once bound to the cell surface, most internalized viral particles trafficked until they reached the appropriate endosomal vesicles to fuse and enter the cytosol, amounting to nearly 80% of endocytosed virions. Our computer-based analysis showed that TOSV Gc shares with RVFV Gc a similar molecular weight and 48% homology in amino acid sequence. As expected, the AlphaFold algorithm predicted that the Gc structure in TOSV resembles RVFV, with an organization typical of a class-II viral fusion protein. Still, further experimental work is needed to solve the X-ray structure of both TOSV Gn and Gc glycoproteins and to determine whether TOSV particles have the same atypical T = 12 symmetry as UUKV and RVFV [8, 9].

From our results, it is clear that TOSV depends for infection on endocytosis and membrane transport within the classical endocytic machinery. After binding, internalization of TOSV occurred, and the virus was observed by confocal microscopy in Rab5a+ EEs. It was efficient and involved about three-quarters of the surface-bound viral particles. The endocytosis step was rapid and completed within 15 min. Expression of Rab5a S34N, which inhibits EE maturation and homotypic EE fusion, reduced TOSV intracellular trafficking and infection. Rab5a Q79L, which provokes expansion of EEs and prevents proper LE maturation, also hampered TOSV infection. One can conclude that TOSV passes through Rab5a-positive EEs, but to be infectious the virus must reach LEs.

Other indications supported the view that TOSV penetrates the cytosol from more acidic downstream organelles. One was the pH threshold for viral membrane fusion below 5.6, another one was the timing of the acid-sensitive step (10–50 min). More directly, confocal microscopy showed that the arrival of virions in Rab7a+ and LAMP1+ LEs coincided with the time of acid activation. Expression of the inactive mutant Rab7a T22N and temperatures below 25˚C impaired TOSV infection. The inhibition of infection after free ubiquitin depletion or MT depolymerization mirrored observations with UUKV, GERV, and unrelated L-PVs, which accumulated in intracellular vesicles and failed to infect [19, 23, 34].

The endosomes provide a milieu in which the decreasing pH is a convenient cue for virus activation [38]. The TOSV-cell fusion demonstrated that low pH is sufficient to trigger fusion. Proteolytic processing in endosomal vacuoles as observed, *e.g.*, for Ebola virus and SARS-CoV-2 was apparently not needed [23, 40]. Acid activation occurred in less than 30 sec, consistent with pH-triggered kinetics observed for other acid-dependent viruses [41]. TOSV most likely follows a fusion process similar to that of RVFV and other phenuiviruses [16], the details of which remain to be elucidated to understand what exactly happens in terms of the structure of Gn and Gc and mechanisms upon acidification and membrane fusion in a cellular context. More functional investigations will be required to determine whether receptors play a role in these mechanisms.

The fusion process was optimal at a pH below 5.5 but also possible at higher pH despite being delayed by ∼100 seconds. Presumably, TOSV can enter the cytosol from endosomes further upstream in late endocytosis pathways, before reaching appropriate LEs for rapid fusion. In this scenario, TOSV could infect tissues and organs devoid of cellular factors necessary for very late virus penetration. The possibility that the virus penetrates earlier may explain the low efficacy of some perturbants of LE maturation in blocking infection by not only TOSV

but also other L-PVs. Typically, drugs such as nocodazole and colcemid interfere with the integrity of MT on which LE depends for maturation [22] but have a weak ability to prevent L-PV infection in general [38]. Nevertheless, whether delayed fusion at suboptimal pH is a specificity of TOSV or a generality among L-PVs remains to be determined.

Another observation supports the notion that infectious entry depends on late endosomal vesicles in their early stages of maturation. The optimal pH for fusion increased from 5.5 to 5.8 when virions were pre-exposed to pH 6.0, a mild acidity typical of EEs. Passage through classical EEs was evidently a necessary step for TOSV to reach LEs. Our data indicate that EEs not only play a role in sorting virions in the degradative pathway of the endocytic machinery but are also important in priming the viral fusion that subsequently occurs in later endosomes. These results suggest that Gn and Gc glycoproteins undergo incremental conformational changes during virion trafficking, *i.e.*, before the membrane fusion process itself. Other phenuiviruses and class-II viruses need to be added to this investigation. However, it is tempting to propose a model in which class-II fusion viruses do not depend solely on a narrow pH threshold for acid-activated penetration, but on the progressive decrease of acidity in endosomes. A model of the TOSV intracellular trafficking and penetration is proposed in Fig 10. In this model, the pH gradient in the endocytosis pathways would activate multiple successive steps in the viral fusion process as virions travel through the endocytic machinery.

From our data, the acidic environment of EEs was clearly important for TOSV entry into the cytosol, but LEs were crucial for viral fusion and infection. TOSV, as well as RVFV, UUKV, and GERV, evidently make atypical use of endosomal acidification in that they can remain infectious in the endocytic machinery long after acid activation. Our results contrast with the widely-accepted view that activation and priming of class-II membrane fusion proteins are irreversible steps and that these fusion proteins act only once [29]. The fusion of phenuiviruses and some other bunyaviruses may specifically involve intermediate steps that are fully or partially reversible; similar mechanisms have been described for class-III fusion proteins [42]. The process likely involves other factors than the Gc glycoprotein, such as Gn on the particle surface or proteins and lipids in cellular target membranes. Although probably not a universal approach, as illustrated by the acid-inactivation of SFV, we cannot exclude that some unrelated class-II fusion proteins follow the same strategy. Reports of class-II fusion proteins are essentially based on structural approaches and out-of-cell context assays, whereas TOSV and other bunyaviruses allowed us the real-time analysis of viral fusion only minutes after virion binding and uptake.

Acid-activated membrane fusion and late penetration appear to be features shared by many phenuiviruses and other bunyaviruses [7, 43, 44]. Our results indicate that TOSV differs from other phenuiviruses in that its penetration seems to rely on LEs in their early stages of maturation, whereas RVFV and UUKV must reach later endosomal compartments and possibly endolysosomes [19]. TOSV showed great resistance to the degradative branch of the endocytic machinery, remaining infectious for a long time after internalization. Overall, the atypical fusion properties of TOSV and other bunyaviruses certainly confer to these viruses a remarkable adaptability to endosomal acidity and the advantage of trial and error in endocytosis pathways until they reach endosomes suitable for viral fusion, the detailed structural biology of which remains a challenge for future work. As such, bunyaviruses have likely found a way to expand their possibilities of entering and infecting host cells, and in turn, facilitating their propagation.

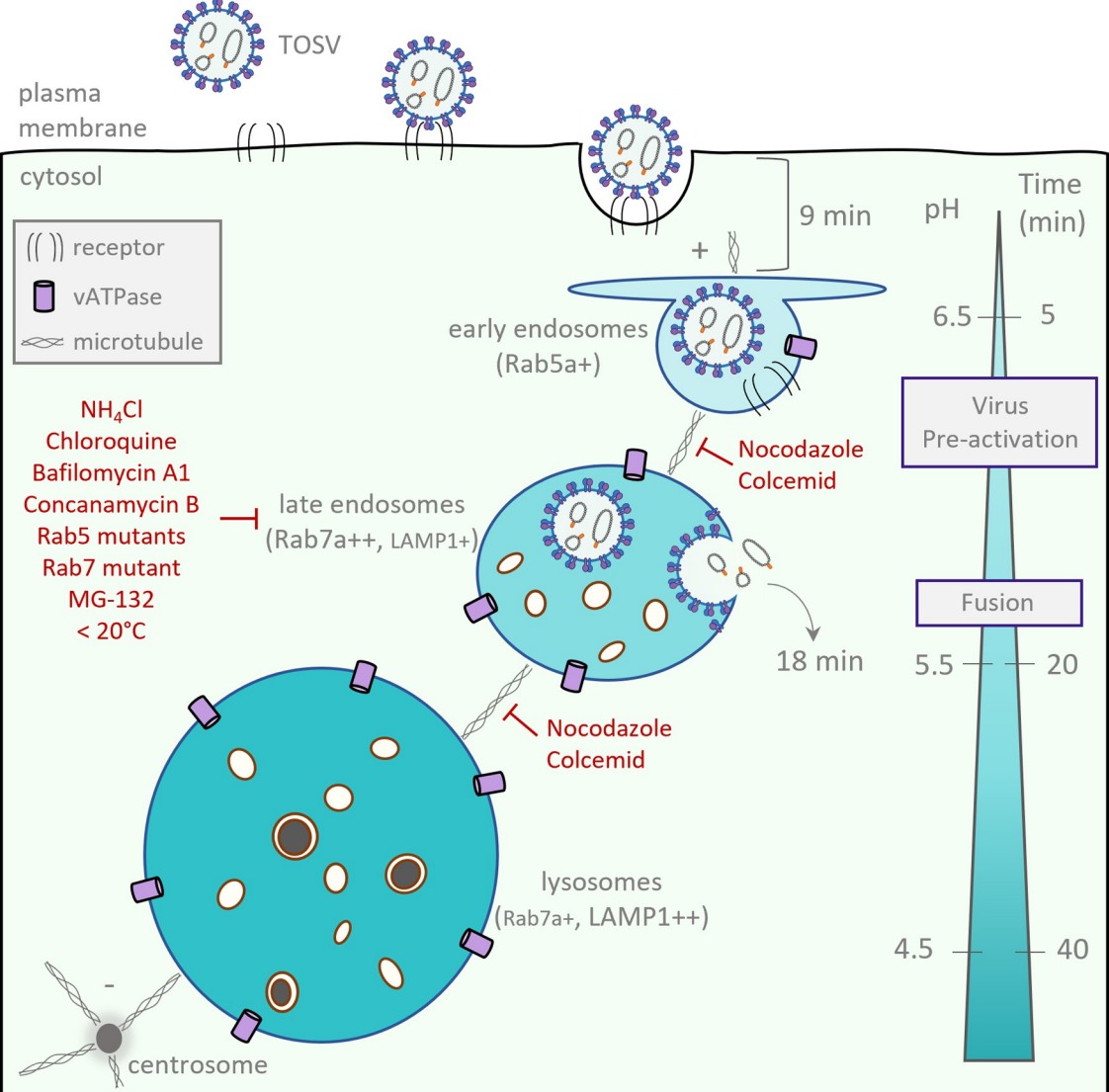

**Fig 10. Entry of Toscana virus (TOSV) into cells.** The time and pH scale on the right side illustrates the typical acidity and timing of cellular cargoes to reach the respective endosomal compartments. +/++ indicate the amount of Rab7a and LAMP-1.

## Materials and methods

### Cell lines

All reagents used for cell culture were obtained from Thermo Fisher Scientific or Merck. Briefly, human A549 and HeLa epithelial cells, HEK293T embryonic kidney cells, Huh-7 hepatocellular carcinoma cells, and U87 glioblastoma cells, as well as chicken DF-1 embryonic fibroblast cells, murine L929 fibroblast cells, canine MDCK kidney epithelial cells, and African green monkey Vero kidney epithelial cells, were grown in Dulbecco's modified Eagle's medium (DMEM) supplemented with 10% fetal bovine serum (FBS). In addition, 1X non-essential amino acids were added to the culture medium of A549 cells. Baby hamster kidney BHK-21 cells were cultured in Glasgow's minimal essential medium (GMEM) supplemented with 10% tryptose phosphate broth and 5% FBS. Human Jurkat and SUP-T1 T lymphoblast cells, raji B cells, and THP-1 monocyte cells were grown in Roswell Park Memorial Institute

(RPMI 1640) containing 10% FBS and the human SH-SY5Y neuroblast cell line in Minimum Essential Medium (MEM)/F12 (Ham's F12) supplemented with 10% serum. The sand fly cells LLE/LULS40 and LLE/LULS45 were derived from embryos of *Lutzomyia longipalpis* and PPL/LULS49 from *Phlebotomus papatasi*. All sand fly cells were cultured in an L-15-based medium in sealed, flat-sided tubes (Nunc) in ambient air at 28°C as reported elsewhere [45, 46]. All cell lines were grown in the presence of 100 units.mL$^{-1}$ penicillin and 100 mg.mL$^{-1}$ streptomycin.

### iPSC-derived neurons

Human induced pluripotent stem cells (iPSCs) derived from a healthy donor (HD6, Heidelberg University) were cultured on Matrigel-coated (Corning) dishes in mTeSR plus medium (STEMCELL Technologies) at 37°C with 5% $CO_2$. Cells were split after 3–5 d, depending on colony size, using EDTA (Sigma). Colonies were scraped off and transferred to Matrigel-coated dishes. The medium was changed every other day. Primary mouse glial cells were prepared as described by Patzke and colleagues [47]. Briefly, newborn (p0) mouse cortices were isolated and digested with papain for 20 min, cells were dissociated by trituration using a thin pipette tip and passed through a cell strainer. Cells were then plated onto T75 flasks in DMEM supplemented with 10% FBS. Upon reaching confluence, glial cells were trypsinized and reseeded twice to remove potential trace amounts of mouse neurons before the glial cell cultures were used for co-culture with induced neuron cells. All procedures involving animals were approved by the Governmental Council Karlsruhe, Germany, and were carried out in strict compliance with German Animal Protection Law (TierSCHG) at the Heidelberg University, Germany. Induced human glutamatergic neurons were generated from HD6 iPSCs as previously described [17]. Briefly, iPSCs were treated with Accutase (Sigma), plated, and simultaneously infected with two lentiviruses: one designed to express rtTA driven by the ubiquitin promoter and another one designed to express, in an inducible manner, NGN2 and puromycin driven by the rtTA promoter. One day later, doxycycline was added to the medium at a concentration of 2 μg.mL$^{-1}$ to drive NGN2 and puromycin expression. Two days later, 1 μg. mL$^{-1}$ puromycin was added to the medium during 24h for selection. After selection, the remaining cells were detached with Accutase and re-plated on Matrigel-coated coverslips along with mouse glia. Half of the medium was then changed every second day for eight days, and 2.5% FBS was added to support astrocyte viability. After day 10, induced neurons were cultured in B27/Neurobasal medium containing Glutamax (Gibco) and 5% FBS for a minimum of 21 days before infection with TOSV.

### Viruses

TOSV strain H4906 (lineage B) [5], recombinant RVFVΔNSs:EGFP [48, 49], GERV [34], UUKV strain S23 [50], SFV [32] and IAV strain PR/8/34 [51] have all been described previously.

### Antibodies

Polyclonal antibody against TOSV structural proteins N, Gn, and Gc was a generous gift from R.B. Tesh (University of Texas, Galveston, Texas, USA) [13]. The mouse monoclonal antibody (mAb) against SFV glycoprotein E2 was kindly provided by Prof. Margaret Kielian (Albert Einstein College of Medicine, USA). The polyclonal guinea pig antibody GR1 against N, Gn, and Gc structural proteins of GERV was recently described [34]. The mouse mAb 8B11A3, which targets a linear epitope in the UUKV nucleoprotein N, is a kind gift from Ludwig Institute for Cancer Research (Stockholm, Sweden) [19]. The mouse mAb that detects the IAV

nucleoprotein was purchased from Merck (MAB8257). Secondary antibodies conjugated to AF405 and AF488 were purchased from Molecular Probes.

## Plasmids and reagents

The plasmids encoding Rab5a, Rab7a, and LAMP1 wt and mutant molecules tagged with EGFP have all been described previously [19]. Stock solutions of chloroquine diphosphate (Sigma) and ammonium chloride ($NH_4Cl$, Sigma) were prepared in $dH_2O$ at concentrations of 19 mM and 1 M, respectively. Bafilomycin A1 (BioViotica), colcemid (Cayman Chemical), concanamycin B (BioViotica), MG-132 (Selleck Chemicals), nocodazole (Merck), and taxol (Merck) were all dissolved in 100% dimethyl sulfoxide (DMSO) to prepare stock solutions at 100 μM, 10 mM, 50 μM, 40 mM, 20 mM, and 10 mM, respectively. Stock solutions of all drugs were diluted in DMEM at the indicated doses (Figs 4 and 5), which are known not to cause cytotoxicity [23, 34]. Both $dH_2O$ and DMSO were included as solvent controls. The hydroxy-succinimidyl (NHS) ester dyes AF488 (Thermo Fisher Scientific) and ATTO647N (Atto-Tec) were dissolved in DMSO (10 and 5 mg.mL$^{-1}$, respectively) while octadecyl rhodamine B chloride (R18, Thermo Fisher Scientific) was dissolved in ethanol (10 mM).

## Virus production, labeling, purification, and titration

TOSV, GERV, UUKV, and SFV were produced in BHK-21 cells in serum-free medium whereas RVFVΔNSs:EGFP was produced in Vero cells in 2%-containing medium and IAV in MDCK cells in serum-free medium [13, 23, 34, 51]. All viruses were purified through a sucrose cushion and then titrated. TOSV, GERV, UUKV, and SFV were titrated on BHK-21 cells and IAV on MDCK cells using a pfu assay following procedures established in the laboratory [51, 52]. The titer of RVFVΔNSs:EGFP was determined in A549 cells by quantifying EGFP-positive cells 7 hpi by flow cytometry using a protocol derived from the approach developed by Barrigua and colleagues [53]. The MOIs are therefore given according to the titers determined in BHK-21 cells for TOSV, GERV, UUKV, and SFV, in A549 cells for RVFVΔNSs:EGFP, and in MDCK cells for IAV. TOSV was fluorescently labeled using a previously-described method [52] in which one and two molecules of AF488 and ATTO647N NHS ester dye, respectively, are conjugated to one molecule of the viral glycoproteins for one of the viral glycoproteins. UUKV was labeled following the same method but with three molecules of AF647 [14]. Alternatively, TOSV ($3x10^9$ pfu.mL$^{-1}$) was labeled with the lipophilic dye R18 (25 μM) [52].

## Virus binding and internalization

Virions were allowed to bind to pre-cooled cells in DMEM containing 0.2% BSA and 20 mM 4-(2-hydroxyethyl)-1-piperazineethanesulfonic acid (HEPES) at pH ∼7.4 (binding buffer) on ice for 90 min at indicated MOIs. Where indicated, virions were first buffered to different pH values in 100 mM citric acid (pH < 5.5), 2-(N-morpholino)-ethanesulfonic acid (MES) (5.5 < pH < 6.5), or HEPES (6.5 < pH < 7.4) for 5 min at 37°C before being returned to pH ∼7.4 and allowed to bind to cells. For internalization assays, cells were rapidly warmed to 37°C and incubated for the indicated periods. Both virus binding and internalization were analyzed by flow cytometry, confocal microscopy, and fluorimetry as described below. To discriminate between internalized and external virions, trypan blue (Sigma) was added to a concentration of 0.01% before the analysis. In flow cytometry- and fluorimetry-based assays, cells were detached from culture plastic by incubation with 0.5 mM EDTA, and virus binding and internalization were performed with cells in suspension in phenol-free binding buffer. In binding competition experiments, cells were first exposed to the indicated amounts of unlabeled

TOSV or SFV for 45 min at 4°C and then to AF488-TOSV at a concentration of 15 nM viral glycoproteins for an additional hour in the presence of unlabeled viruses on ice.

## Infection assay

Cells were exposed to viral particles at the indicated MOIs in the respective medium without serum for 1 h at 37°C. Virus inoculum was then replaced by the respective complete medium, and infection was allowed for an additional 5 h for TOSV and 7 h for RVFV, GERV, UUKV, SFV, and IAV, if not indicated otherwise. Where mentioned, virions were buffered to the indicated pH values as described in the Virus binding and internalization section before the infection of cells. For inhibition experiments, cells were pretreated for 30 min at 37°C, except for colcemid, nocodazole, and taxol. For these three drugs, cells were exposed for 3 h at 4°C before infection was carried out at 37°C. For all the drugs, cells were infected in the continuous presence of the inhibitors. To assess the dependence of TOSV entry on temperature, virus entry was first synchronized on ice. Infected cells were then incubated for 50 min at the indicated temperatures before warming to 37°C in the respective complete medium containing 50 mM $NH_4Cl$ and buffered with 20 mM HEPES for 6 h. For $NH_4Cl$ add-in time courses, virus binding to cells was synchronized on ice. Cells were then rapidly warmed to 37°C, and $NH_4Cl$ (50 mM) was added at the indicated times. Cells were subsequently incubated at 37°C and harvested 6–8 h after the initial warm shift. In the reverse approach, virus binding to cells was synchronized on ice, and cells rapidly warmed to 37°C in the presence of $NH_4Cl$ (50 mM) before the weak base was washed out at the indicated times. Cells were then incubated for an additional 6 h at 37°C. Virus infection was monitored by flow cytometry.

## Flow cytometry

Infection was monitored by flow cytometry as previously described [19]. Briefly, infected cells were fixed and permeabilized by 0.1% saponin before the immunofluorescence staining of newly-produced viral proteins with respective primary antibodies against TOSV, GERV, UUKV, SFV, or IAV at concentrations of 2.5 $\mu g.mL^{-1}$, 1:16,000, 1:1,000, 1:400, and 1:250, respectively. Cells were then washed and exposed to anti-guinea pig or anti-mouse AF405- or AF488-conjugated secondary antibodies (1:500, Thermo Fisher Scientific) for one hour. Alternatively, cells infected with RVFVΔNSs: EGFP were assayed for the EGFP signal, and in binding and internalization experiments, AF488-labeled virions were directly measured. Infection was quantified with a Celesta flow cytometer (Becton Dickinson) and FlowJo software v10.6.2 (TreeStar).

## Viral protein analysis

Purified virus stocks were diluted in lithium dodecyl sulfate (LDS) sample buffer (Thermo Fisher Scientific) and separated by SDS-PAGE (NuPAGE Novex 10% Bis-Tris gel, Thermo Fisher Scientific) as previously described [5]. Viral proteins were either stained with Coomassie blue or analyzed by fluorography using an LI-COR Odyssey CLx scanner and ImageJ v1.53c [National Institute of Health (NIH), US].

## Cryo-electron microscopy

Sucrose gradient-purified virus particles were washed in a buffer containing 10 mM HEPES, 150 mM NaCl, 1 mM EDTA, pH ∼7.3, pelleted by ultracentrifugation, and fixed with 4% paraformaldehyde. Subsequently, 2.5 μL of the fixed virion solution was applied to degassed Quantifoil R2/2 Cu grids that were discharged at 30 mA for 2 min before sample application. The

sample was vitrified in liquid ethane using a Leica EM GP2 plunge freezer at 4˚C and 90% humidity, and sensor blotting from the reverse side for 3 sec. Data were acquired using SerialEM software on a Thermo Fisher Scientific Glacios transmission electron microscope operated at 200 kV and equipped with a Falcon 3 direct electron detector. Before data acquisition, the microscope was adjusted by a comma-free alignment in SerialEM, and the gain reference was determined. Regions of interest were identified in low-magnification setups. For high-resolution data acquisition, the nominal magnification was 73,000, resulting in a pixel spacing of 2.019 Å. The camera was operated in linear mode with a dose rate of 16 e-/s/pixel. The total dose was 19.6 e-/$\text{Å}^2$ and was divided into 22 dose-fractions that were aligned and gain-corrected in SerialEM. Cryo-EM micrographs were analyzed using ImageJ v1.53c (NIH). The length and width of a viral particle were determined by measuring the largest and smallest distances between peaks in density profiles or membranes on the opposite side of the viral particle.

## Fluorescence microscopy

Cells that were exposed to fluorescently labeled virions were mounted with Mowiol (Merck), and if indicated, nuclei were stained with Hoechst 33258 (0.5 µg.mL$^{-1}$, Thermo Fisher Scientific). Live cell imaging was performed in the continuous presence of viral particles. Both fixed and live samples were imaged with a Leica TCS SP8 confocal microscope equipped with an HC PL APO CS2 63x/1.4 N.A. oil immersion objective. In addition, super-resolution microscopy was used to image ATTO647N-TOSV mounted in Mowiol on PEI-coated coverslips with a 2-color-STED microscope (Abberior instruments GmbH) as described by Kummer and colleagues [51]. The STED microscope was equipped with an x100 Olympus UPlanSApo (NA 1.4) oil immersion objective, and the pixel size was set to 60 nm (confocal) and 15 nm (non-diffracted), respectively. Minor contrast and brightness adjustments of images and Richardson–Lucy deconvolution (regularization parameter of $10^{-3}$, stopped after 30 iterations) were carried out using Imspector software 16.1.7098 (Abberior instruments). Images were analyzed with ImageJ v1.53c software (NIH) and the Imspector software (Abberior Instruments GmbH).

## DNA transfection

A549 cells ($8\text{x}10^4$) were transfected with 500 ng of plasmids encoding Rab5a, Rab7a, and LAMP1 wt and mutant molecules tagged with EGFP using Lipofectamine 2000 (Thermo Fisher Scientific) according to the manufacturer's recommendations. Supernatants were replaced by fresh medium 5 h after transfection, and the cells were incubated for an additional 17 h before exposure to TOSV.

## Flow cytometry-based plasma membrane virus fusion

TOSV was induced to fuse with the plasma membrane as previously described [24]. Briefly, TOSV binding to cells at the indicated MOIs was synchronized on ice, and cells were subsequently exposed to buffers of different pH values as indicated for 90 sec at 37˚C. Infected cells were then washed extensively and incubated in a complete medium at pH $\sim 7.4$ supplemented with $NH_4Cl$ (50 mM) for 7 h. Infection was quantified by flow cytometry following the immunostaining of TOSV structural proteins.

## R18-based virus fusion

The fusion of R18-TOSV with host cell membranes was performed as previously described [34]. Briefly, cells were detached from the culture surface using 0.5 mM EDTA, and binding of

R18-TOSV at MOI 10 to cells in suspension was synchronized on ice in a phenol-free medium at pH ∼7.4 for 90 min. To determine the kinetics of virus penetration, virus-bound cells were rapidly warmed in an FP-8500 fluorometer (Jasco) to 37˚C, and the emission of fluorescence was measured over 90 min. For virus fusion with cell membranes, virus-bound cells were rapidly warmed inside the fluorometer to 37˚C, and the fusion was triggered with buffers of varying pHs as indicated. The fluorescence was measured for 600 sec. Where indicated, virions were pre-exposed to buffers at pH ∼7.4 or 6.0 as described in the Virus binding and internalization section.

## Structural modeling of TOSV Gc

The amino acid sequences of the M segment of TOSV and RVFV, strains H4906 and 35/74, respectively, were first aligned and analyzed with EMBOSS Needle or blastp suite-2sequences using a BLOSUM62 matrix and Multiple Sequence Alignment Viewer 1.22.2. ColabFold v1.5.2, an algorithm that combines MMseqs2 with AlphaFold2 [54], was then used to predict the structure of TOSV Gc strain H4906 in pre- and post-fusion conformation. For this analysis, the default settings were applied, and the pre- and post-fusion structures of RVFV (PDB, 6F9F [16], and PDB, 6EGT [30], respectively) served as models. The AlphaFold predictions for TOSV Gc were visualized with the PyMOL Molecular Graphics System, v2.5.4 Schrödinger, LLC. Structural comparisons with RVFV Gc were achieved using UCSF ChimeraX, a "matchmaker" plugin to align the models, and a "morph" plugin to generate a conformation change trajectory [55]. PDB files of TOSV Gc pre- and post-fusion models are available upon request.

## Statistical analysis

Graph plotting and statistics were achieved with Prism v8.0.1 (GraphPad Software). The data shown are representative of at least three independent experiments. Values are presented as the means of triplicate experiments ± standard deviation if not stated differently.

## Supporting information

**S1 Fig. Quantification of Toscana virus (TOSV) infection in A549 cells.** A549 cells were exposed to TOSV at MOI 1 for up to 24 h. Cells were then fixed and permeabilized, and infection was monitored by flow cytometry after immunostaining against all TOSV structural proteins, *i.e.*, N, Gn, and Gc.
(TIF)

**S2 Fig. Fluorescence labeling of Toscana virus (TOSV).** (A) The picture shows a linear sucrose gradient after ultracentrifugation with unbound ATTO647N dye on the top and a band that corresponds to ATTO647N-TOSV particles at a density between 40 and 45% sucrose. (B) Fluorescent particles (ATTO647N-TOSV) and unlabeled TOSV were analyzed by nonreducing SDS-PAGE with fluorography (right panel) and then Coomassie blue staining (left panel). (C) ATTO647N-TOSV particles were imaged by confocal microscopy (top panel) and STED microscopy (bottom panel). Scale bar, 1 μm. (D) Fluorescently labeled TOSV particles were analyzed by the pfu assay shown in Fig 1D, and the titers were normalized to the amount of the viral nucleoprotein N.
(TIF)

**S3 Fig. Sequence alignment of TOSV and RVFV M segments.** The M segment of TOSV H4906 strain and RVFV 35/74 strain were aligned with EMBOSS Needle. Identical amino acids are highlighted by '|', similar amino acids by ':', and different amino acids '.'. The sequence in the M segment corresponding to NSm, Gn, and Gc were highlighted in blue,

green, and yellow, respectively.
(PDF)

**S1 Table. Susceptibility of different mammalian and sand fly cell lines to Toscana virus (TOSV).** [a]Cells were infected with TOSV at MOI 1 for 18 h, fixed, permeabilized, and immunofluorescently stained against all TOSV structural proteins. Infection was quantified by flow cytometry, and the sensitivity of cells to TOSV infection (percentage of infected cells) was given as follows: +++ greater than 30%, ++ from 10% to 30%, + from 1% to 10%, - less than 1%. [b]The production of viral progeny was assessed by pfu titration assay and is given according to the size of plaques 72 hpi as follows: +++ greater than 1 mm, ++ from 0.5 to 1 mm, + less than 0.5 mm, - no plaques. n.d., not determined.
(PDF)

**S1 Movie. Coordinated motion of Toscana virus (TOSV) with Rab5a+ endosomal vesicles.** A549 cells stably expressing EGFP-Rab5a were exposed to ATTO647N-TOSV at MOI 10 and imaged every 15 sec at 37˚C by confocal microscopy for 25 min. TOSV particles (magenta) are seen moving with EGFP-Rab5a+ vesicles (green) approximately 11 min post-warming.
(AVI)

**S2 Movie. Coordinated motion of Toscana virus (TOSV) with Rab7a+ endosomal vesicles.** A549 cells stably expressing EGFP-Rab7a were exposed to ATTO647N-TOSV at MOI 10 and imaged every 15 sec at 37˚C by confocal microscopy for 25 min. TOSV particles (magenta) are seen moving with EGFP-Rab7a+ vacuoles (green) approximately 24 min post-warming.
(AVI)

**S3 Movie. Conformational transition from the pre- to the post-fusion state of Toscana virus (TOSV) Gc.** The structural models of the pre- and post-fusion ectodomains predicted with ColabFold in Fig 7B and 7C were first adjusted for position, and 120 images were then generated to obtain the morph trajectory using UCSF ChimeraX and the "morph" plugin with the wrap parameter set as true. The final movie was recorded at 25 frames per second.
(MP4)

**S1 File. Protein Data Bank (PDB) file describing the three-dimensional structure of the Toscana virus glycoprotein Gc in its pre-fusion state predicted by AlphaFold2.**
(PDB)

**S2 File. Protein Data Bank (PDB) file describing the three-dimensional structure of the Toscana virus glycoprotein Gc in its post-fusion state predicted by AlphaFold2.**
(PDB)

**S3 File. This Excel workbook includes all the numerical values used to produce the graphs shown in this study.**
(XLSX)

## Acknowledgments

We acknowledge Elodie Chatre and the Imaging Platform Platim, SFR Biosciences, Lyon, as well as Vibor Laketa and the Infectious Diseases Imaging Platform (IDIP) at the Center for Integrative Infectious Disease Research (CIID) Heidelberg. The sand fly cell lines were supplied by the Tick Cell Biobank at the University of Liverpool. F.K.M.S. acknowledges support from the Scientific Service Units (SSUs) of ISTA through resources provided by the Electron Microscopy Facility (EMF).

## Author Contributions

**Conceptualization:** Jana Koch, Pierre-Yves Lozach.

**Formal analysis:** Jana Koch, Qilin Xin, Martin Obr, Susann Kummer.

**Funding acquisition:** Lesley Bell-Sakyi, Florian KM Schur, Claudio Acuna, Pierre-Yves Lozach.

**Investigation:** Jana Koch, Martin Obr, Alicia Schäfer, Nina Rolfs, Holda A. Anagho, Aiste Kudulyte, Lea Woltereck.

**Methodology:** Jana Koch, Qilin Xin, Martin Obr, Susann Kummer, Joaquin Campos.

**Project administration:** Pierre-Yves Lozach.

**Supervision:** Jana Koch, Zina M. Uckeley, Hans-Georg Kräusslich, Florian KM Schur, Claudio Acuna, Pierre-Yves Lozach.

**Validation:** Jana Koch.

**Visualization:** Jana Koch, Qilin Xin, Pierre-Yves Lozach.

**Writing – original draft:** Jana Koch, Pierre-Yves Lozach.

**Writing – review & editing:** Jana Koch, Qilin Xin, Martin Obr, Alicia Schäfer, Holda A. Anagho, Susann Kummer, Zina M. Uckeley, Lesley Bell-Sakyi, Florian KM Schur, Claudio Acuna, Pierre-Yves Lozach.

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
