## [Decision Letter · Decision Letter 0]

2 May 2023

Dear Dr. Lozach,

Thank you very much for submitting your manuscript "The phenuivirus Toscana virus makes an atypical use of vacuolar acidity to enter host cells" for consideration at PLOS Pathogens. As with all papers reviewed by the journal, your manuscript was reviewed by members of the editorial board and by several independent reviewers. In light of the reviews (below this email), we would like to invite the resubmission of a significantly-revised version that takes into account the reviewers' comments. We would ask the authors to carefully address the comments by reviewer 1 on what is considered to be atypical for TOSV, eg different from other bunyavirus entry processes, as this seems unclear in the manuscript.

We cannot make any decision about publication until we have seen the revised manuscript and your response to the reviewers' comments. Your revised manuscript is also likely to be sent to reviewers for further evaluation.

Sincerely,

Alain Kohl

Academic Editor

PLOS Pathogens

Kanta Subbarao

Section Editor

PLOS Pathogens

Kasturi Haldar

Editor-in-Chief

PLOS Pathogens

orcid.org/0000-0001-5065-158X

Michael Malim

Editor-in-Chief

PLOS Pathogens

orcid.org/0000-0002-7699-2064

We would ask the authors to specifically also address the comments by reviewer 1 on what is considered to be atypical, eg different from other bunyavirus entry processes.

Reviewer's Responses to Questions

**Part I - Summary**

Reviewer #1: This study examines the pathway by which Toscana virus (TOV) infects cell lines. TOV is a Bunyavirus though it has not been studied nearly as much as some other members of this virus family such as Rift Valley Fever virus. Bunyaviruses are know to be endocytosed after binding to the cell surface and are delivered to endosomes where the low pH environment induces the viral glycoproteins to under conformational changes that elicit fusion between the viral and endosomal membranes, allowing the viral genome to gain entry into the cytoplasm. These viruses have what are termed Class II fusion proteins, a number of which have been very well studied.

This is a technically strong, very detailed study in which the investigators use pretty much every experimental approach that has been used to study viral cell entry pathways, from very classic approaches such as using lysosomotropic agents to elevate vacuolar pH to membrane fusion assays using fluorescent dyes and more modern assays in which mutant proteins (such as a form of Rab5) are expressed to as to interdict discrete steps in the endocytic pathway. Everything worked as expected and the results are clear and consistent: TOV like other Bunyaviruses is delivered to endosomes and undergoes low-pH dependent membrane fusion. A few comments follow.

• I don’t understand the use of ‘atypical’ in the manuscript title. I didn’t see anything that was unexpected or atypical in this series of very nice experiments. Concerned that I had missed something, I did a search for ‘atypical’ in the document, and other than stating that the virus makes an atypical use of vacuolar acidity, the word is not used in the results at all. The title made me think that there was something different about how this viruses enters cells, but such was not the case.

• If the paper had to be trimmed, I would suggest really cutting back or even eliminating the morphological studies with labeled virions. The problem as I suspect the investigators know if the infectious particle ratio: at best it seems only 1 in 10 virions are infectious, so when looking a labeled particles by microscopy one is basically looking at non-infectious viruses that may or may not be doing the same thing as infectious viruses at early stages of infection. The experiments that immediately follow this (using inhibitors such as Rab7a) are more definitive.

• I don’t think I would use the term ‘rather inefficient’ in describing binding to TOV to A549 cells (about 25% bound under the conditions used). I actually thought that this was rather efficient binding, but the issue is that there is no basis for comparison. Unless if the authors want to look at binding of other virions to these cells under the same conditions, I would argue that the use of subjective terms in this setting is perhaps not appropriate.

• The acid-bypass experiments shown in Figure 5 are, I think, the correct control for the lysosomotropic experiments shown in Figure 1. At first I thought this control was missed, but it appeared later. I’ld probably re-organize things, but that is just one person’s opinion and so I would just ask the authors to consider it.

Reviewer #2: The authors report new elements concerning the mechanisms of entry and fusion of the Toscana Virus in iPSC-derived neurons and in different cell lines. The manuscript is interesting and several experiments have been carried out to provide a valid hypothesis on how binding internalization and membrane fusion can occur during TOSV infection.

The rationale of this study is clear; the methodology pursued in this study is accurate and well-conducted. The results are helpful to understand the mechanisms by which TOSV, as well as other members of Phleboviridae, enter and infect host cells.

Reviewer #3: Koch et al. report an extensive and well-executed characterization of Toscana virus (TOSV) entry, providing alongside the data on virus morphology and predicted structural model for the fusion Gc protein. The authors developed flow cytometry, fluorescence microscopy and fluorimetry assays to characterize different stages of TOSV entry, integrating the data into a comprehensive TOSV entry model with several key points: only a small fraction of virus (�20%) efficiently binds to cells, majority of internalized virions fuse within the endosomal compartment (�80%), the fusion process is optimal at pH of 5.5 and requires passage of the virions from the early to the late endosomal (EE, LE, respectively) compartment, where the membrane fusion and release into cytosol occurs. The authors report a new and interesting observation that TOSV and two more bunyaviruses remain infectious after exposure to low pH, in contrast to most fusion proteins where the low pH triggers an irreversible conformational change of the fusion protein, rendering such virions inactivated.

The novelty of this manuscript relies in the studied object, the TOSV, which is a re-emerging human pathogen in south Europe and north Africa, which has been poorly investigated and characterized. The findings presented here therefore fill an important gap in knowledge related to the virus that is responsible for a growing number of neuro-invasive infections that lead to meningitis in most severe cases. The data in the manuscript are presented in a clear and logical way (with the exception of some figures, see below), and the experiments are exhaustive, well-thought and well-described.

**Part II – Major Issues: Key Experiments Required for Acceptance**

Reviewer #1: Everything is above

Reviewer #2: It is not mentioned that the virus (TOSV) was plaque-purified, and it was collected from BHK-21 infected cells. The fact that most of the infecting virus is not infectious could be due to the presence of a lot of defective particles present in the viral stock. Although it is known that virions can have even less than 3 genomic RNA segments, it would be useful to work with a rather pure viral stock. The virus was purified on sucrose gradient, but the band could be contaminated with defective particles.

Discussion: the results of the experiments are sometimes reported in a not well organized way, therefore the reader may lose the thread. This is perhaps due to the amount of data the authors need to comment on. It appears that the acidic environment of the EEs is useful for the entry of TOSV into the cytosol, but LES are crucial for infection, therefore the role of both Endosomes should be better explained. I would suggest to make it clearer, trying to better schematize the path of the virus inside the cell until fusion.

Reviewer #3: 1. A figure that summarizes the findings and represents the proposed entry model, indicating the timing of events (maybe on a separate panel) would be very useful to the readers.

2. Figure legends need to be more succinct and to contain only the relevant methods info. Most of the figure legend text in the current version would be more appropriate for MM.

3. Lines 180-181: “Time-course analysis of the generation of trypan blue-resistant fluorescence of cell-associated TOSV-AF488 revealed that internalization into A549 cells started within the first 5 min and increased over time to reach the half-maximal level (t1/2) within 9+/-2 min and the plateau 10 min later (Fig. 2I).” While the internalization between minutes 20-40 does not increase at the same rate as during first 20 minutes, the number of data points on Figure 2I are not sufficient to support the statement that the plateau is reached at 20 minutes. They rather indicate a slower increase.

4. Figure 6 – the sequence alignment on panel A should be included in the SI using the font size that can be read without magnifications. The choice of yellow color for RVFV on panels D and E is not the best one. I would suggest grey, or another color that stands out better on the white background. On panels D and E it would be helpful to have the two structures next to each other in addition to the superposition, because it is difficult to visualize the described rotation. Domains I, II and III should be labeled on panels D and E, and a magnified view at domain I to depict where the strands A0B0 and C0D0 located, and how they are different from the RSFV.

5. Could the authors provide an explanation as to why TOSV infections of AD549 (Figure 7A) vs BHK21 cells (Figure 7D) show different trends – more or less around 100% in A549 cells at all pH values, and going from 20% at pH 5 to 100% at pH 7.5 in BHK cells? Also, was it possible to infect A549 cells (instead of BHK21) with SFV and UUKV to have a fairer comparison?

**Part III – Minor Issues: Editorial and Data Presentation Modifications**

Reviewer #1: Everything is above. I wrote my review before seeing that you have a preferred format. Sorry!

Reviewer #2: - Besides 1 and 2, there are more specific references to be added describing most severe cases of neurologic infections by TOSV.

-Lines 94: it would be better to specify the three sandfly cell lines used, although mentioned in Mat & Meth.

-Lines 239-240: ‘Additional processing…’ is a little bit confusing for the reader, since the glycoproteins have a role in the fusion.

Line 474: please write the full form of the acronym ‘L-PVs’, as it is mentioned for the first time, and use the acronym from then on.

- Is Ref. 53 missing?

Reviewer #3: 1. Figure 1 – could the graphs on C and E panels be grouped into a single panel that contains the label ‘neurons’ to designate the type of cells used, while the graphs on D and F could similarly be grouped into a single panel with a label ‘A549 cells’

2. Figure 2 – the white dots are really hard to see on the printed version of the Figure. Could the panels D and F be enlarged or contrast chosen to show the dots better?

3. Figure 3 – adding labels ‘EE’, ‘LE’, and ‘Lysosome’ to panels B, D, and F, respectively, would orient the reader who is not familiar with the markers of different endosomal compartments (or include that info in the figure legend?)

4. Figure 4 – this figure is very difficult to follow. I suggest grouping the panels by the type of cells or consolidating the panels by some other criteria. For example, graphs D, F, and H could be in the same panel (neurons), and graphs E, G, I and J could be in another panel (A594 cells).

5. Figure 5 – there is no call for panel A in the manuscript. The titles of Y axes on panels B, C, E and F should not include the (% of XX min / C) – this is confusing and could be explained in the legend. Or it could be changed into “% of the infected cells relative to the 80 min data point” (which becomes too long…).

6. Figure 7 legend: Could the description of panels B-E be changed to be easier to follow? Alternatively label the type of cell and virus directly on the graph and skip the explanation in the figure legend

7. Line 143 – it should be noted how the measured length of the protrusions compares to the length of the predicted pre-fusion Gc (figure 6C).

8. Lines 150-151: “The unbound virions were washed away, and the remaining cell associated, fluorescent viral particles were determined with a fluorimeter” – the sentence is missing a full subject. What was determined – the amount, number, % of fluorescent viral particles…?

9. Lines 225-226: Could the authors mention why taxol was not tested on neurons?

10. Lines 242 – 288: This section needs some polishing to improve the clarity.

a. For example, line 246 “Virus entry was then synchronized by rapid warming...” – could it be explained why this synchronizes the virus?

b. Line 248 – “…concentration of 50 mM was used to ensure that infection …” – instead say “NH4Cl was added to 50mM to ensure …”.

c. Figure 5A is not mentioned.

d. Lines 273-275: Calls for figure should be made at appropriate places instead at the end of the sentence. The sentence should be modified to: “At 25 °C, fusion corresponded to 40% of the 37 °C control (Figure 5F), whereas infection via the normal route was lowered by 85% (Figure 5E).

e. Lines 278-298: In the sentence “Most likely the viral particles did not infect at 25 °C and below because they did not reach a compartment with low enough pH.” – does the “compartment with low enough pH” correspond to LE? If yes, why not be specific and use ‘LE’ instead?

11. Lines 321-323: Could the authors elaborate more on this interesting statement reported in reference 30?

12. Line 364 – several time the expression ‘we forced … fusion’ is used in the manuscript. Choice of ‘induced’ instead of ‘forced’ might be better?

PLOS authors have the option to publish the peer review history of their article (what does this mean?). If published, this will include your full peer review and any attached files.

Reviewer #1: No

Reviewer #2: No

Reviewer #3: No

Figure Files:

Data Requirements:

Please note that, as a condition of publication, PLOS' data policy requires that you make available all data used to draw the conclusions outlined in your manuscript. Data must be deposited in an appropriate repository, included within the body of the manuscript, or uploaded as supporting information. This includes all numerical values that were used to generate graphs, histograms etc.. For an example see here on PLOS Biology: http://www.plosbiology.org/article/info:doi%2F10.1371%2Fjournal.pbio.1001908#s5.
---

## [Decision Letter · Decision Letter 1]

17 Jul 2023

Dear Dr. Lozach,

We are pleased to inform you that your manuscript 'The phenuivirus Toscana virus makes an atypical use of vacuolar acidity to enter host cells' has been provisionally accepted for publication in PLOS Pathogens.

Best regards,

Alain Kohl

Academic Editor

PLOS Pathogens

Kanta Subbarao

Section Editor

PLOS Pathogens

Kasturi Haldar

Editor-in-Chief

PLOS Pathogens

orcid.org/0000-0001-5065-158X

Michael Malim

Editor-in-Chief

PLOS Pathogens

orcid.org/0000-0002-7699-2064

Reviewer Comments (if any, and for reference):

Reviewer's Responses to Questions

**Part I - Summary**

Reviewer #1: I think the authors have done a fair job of addressing the points raised by the reviewers.

Reviewer #2: The revised version of the manuscript is clearer and brings new knowledge about Toscana virus, reporting characteristics peculiar to TOSV and common to other bunyaviruses useful to understand their structural biology.

Reviewer #3: The authors have done all the necessary modifications. I have no further comments.

**Part II – Major Issues: Key Experiments Required for Acceptance**

Reviewer #1: (No Response)

Reviewer #2: I do not have major issues to ask

Reviewer #3: No major issues.

**Part III – Minor Issues: Editorial and Data Presentation Modifications**

Reviewer #1: (No Response)

Reviewer #2: I do not have minor issues to ask

Reviewer #3: No minor issues.

PLOS authors have the option to publish the peer review history of their article (what does this mean?). If published, this will include your full peer review and any attached files.

Reviewer #1: No

Reviewer #2: No

Reviewer #3: No

---

## [Editor Report · Acceptance letter]

7 Aug 2023

Dear Dr. Lozach,

We are delighted to inform you that your manuscript, "The phenuivirus Toscana virus makes an atypical use of vacuolar acidity to enter host cells," has been formally accepted for publication in PLOS Pathogens.

Best regards,

Kasturi Haldar

Editor-in-Chief

PLOS Pathogens

orcid.org/0000-0001-5065-158X

Michael Malim

Editor-in-Chief

PLOS Pathogens

orcid.org/0000-0002-7699-2064